# Exploring Non-Gaussian Sea Ice Characteristics via Observing System Simulation Experiments

Christopher Riedel [1] and Jeffrey Anderson [2]

[1]Advance Study Program, National Center for Atmospheric Research, Boulder, Colorado
[2]Data Assimilation Research Section, National Center for Atmospheric Research, Boulder, Colorado

**Correspondence:** Christopher Riedel (criedel@ucar.edu)

**Abstract.** The Arctic is warming at a faster rate compared to the globe on average, commonly referred to as Arctic amplification. Sea ice has been linked to Arctic amplification and gathered attention recently due to the decline in summer sea ice extent. Data assimilation (DA) is the act of combining observations with prior forecasts to obtain a more accurate model state. Sea ice poses a unique challenge for DA because sea ice variables have bounded distributions, leading to non-Gaussian distributions. The non-Gaussian nature violates Gaussian assumptions built into DA algorithms. This study presents different observing system simulation experiments (OSSEs), which through experimental observation networks and synthetic observations will provide a data assimilating testing framework. The OSSEs framework will help determine the best data assimilation configuration for assimilating sea ice and snow observations. Findings indicate that assimilating both sea ice thickness and snow depth observations while omitting sea ice concentration observations produced the best sea ice and snow forecasts, in our idealized experimental setup. A simplified DA experiment helped demonstrate that the DA solution is biased when assimilating sea ice concentration observations. The biased DA solution is related to the observation error distribution being a truncated normal distribution, and the assumed observation likelihood is normal for the DA method. Additional OSSEs show that using a non-Gaussian DA method does not alleviate the non-Gaussian effects of sea ice concentration observations, and assimilating sea ice surface temperatures has a positive impact on snow updates. Finally, it is shown that perturbed sea ice model parameters, used to create additional ensemble spread in the free forecasts, lead to a year-long negative snow volume bias.

## 1 Introduction

Warming over the Arctic region, a phenomenon commonly referred to as Arctic amplification (Serreze and Francis, 2006), has been identified in both observations (Serreze et al., 2009; England et al., 2021) and climate models (Holland and Bitz, 2003). Numerous studies have found this warming rate to be approximately twice as fast as the global average (Walsh, 2014; Jansen et al., 2020; Yu et al., 2021). A recent study found that Arctic amplification-related warming could be three-to-four times faster than the global average, more than double the warming rate previously estimated (Rantanen et al., 2022). Projections of Arctic amplification rely heavily on the ability of coupled numerical models to represent each Earth-system component. One important Earth-system component linked to Arctic amplification–the cryosphere–has gathered attention recently due to the declining summer sea ice extent over the recent decades (Screen and Simmonds, 2010; Jenkins and Dai, 2021). During wintertime, sea

ice can act as an insulator trapping ocean heat, created from the absorbed shortwave radiation during the summer sea-ice loss season, within the ocean allowing for cooler winter atmospheric temperatures (Chung et al., 2021). Additionally, snow cover on top of sea ice can impact seasonal sea ice evolution, growth and melt (Holland et al., 2021). Providing more accurate sea ice and snow states via data assimilation in our coupled Earth-system modeling frameworks could help improve future projections of the climate and the processes related to Arctic amplification.

Data assimilation (DA) is the action of optimally combining information from prior forecasts with observations to improve the current estimate of the state of any Earth-system component. The statistical methods used to optimally combine this information often have Gaussianity assumptions, depending on the choice of the data assimilation method. One data assimilation method that has commonly been applied in Earth-system problems is the ensemble Kalman filter (EnKF; Evensen 2003; Houtekamer and Zhang 2016), which includes Gaussian assumptions in its original Kalman filter formulation (Kalman, 1960). These Gaussian assumptions can lead to biased solutions when prior forecast distributions are non-Gaussian or errors associated with the observations are also non-Gaussian. Common sea ice variables have both double and single bounded quantities (e.g., doubly-bounded: sea ice concentration; singly-bounded: sea ice thickness) that lead to non-Gaussian distributions, which would violate Gaussian assumptions. Studies have investigated the performance of different EnKF formulations (stochastic versus deterministic) under non-Gaussian conditions and found that while the stochastic formulation was more stable, both had biased solutions (Lawson and Hansen, 2004; Lei et al., 2010). Different ensemble data assimilation methods that remove the Gaussian assumption have been proposed, however, many have only been tested in low-order models and could be potentially expensive in high-dimensional geophysical models (Pham, 2001; Anderson, 2010; Sakov et al., 2012b; Metref et al., 2014). Here, instead of testing a new ensemble data assimilation method, we will conduct experiments to highlight the impacts of different non-Gaussian sea ice variables during data assimilation updates.

The application of data assimilation to sea ice problems is not a novel idea since this research topic has been investigated for more than two decades. Common observation descriptive quantities for sea ice are concentration (e.g., the fraction of a grid cell covered with sea ice) and thickness (e.g., the sea ice surface extending down into the ocean). Previous studies have highlighted the importance of initial conditions when trying to predict Arctic sea ice from local to seasonal time scales, especially regarding accurate initialization of sea ice thickness (Msadek et al., 2014; Day et al., 2014; Dirkson et al., 2017). Although different data assimilation techniques have been used to update sea ice state variables (Meier and Maslanik, 2003; Van Woert et al., 2004; Lindsay and Zhang, 2006; Stark et al., 2008), numerous studies have tested updating sea ice state variables using the EnKF data assimilation method (Lisæter et al., 2003; Barth et al., 2015). These EnKF studies were tested both in a synthetic observation framework referred to as observing system simulation experiments (OSSEs; Barth et al. 2015; Kimmritz et al. 2018; Zhang et al. 2018) and using real observations from remote sensing platforms (Sakov et al., 2012a; Massonnet et al., 2015). These studies found improvements in both sea ice analyses and their corresponding forecasts related to the spatial sea ice concentration field but little improvement in sea ice thickness. In addition, studies have improved the initialization of sea ice cover when updating sea ice thickness via a multivariate framework when assimilating only sea ice concentration observations (Massonnet et al., 2015; Sakov et al., 2012a). More recent studies have tested the assimilation of sea ice thickness observations and found further improvements to both sea ice thickness and sea ice concentration states (Mathiot et al., 2012; Chen et al., 2017; Fritzner et al.,

2018; Mu et al., 2018; Fiedler et al., 2022). While results from assimilating sea ice thickness observations are positive, they contain large observation uncertainties because satellite remote sensing retrieval algorithms contain large uncertainties due to input parameters and instrument errors (Kwok and Cunningham, 2008; Zygmuntowska et al., 2014; Tilling et al., 2016; Xie et al., 2016; Ricker et al., 2017). Further research is needed to determine how to properly handle these uncertainties when assimilating sea ice observations. Lastly, there have been recent attempts to obtain observed snow depth from satellites; however, the uncertainties associated with these observations remain high (Maaß et al., 2013; Rostosky et al., 2018). Because snow is closely connected to albedo and sea ice melting, further understanding of the impacts of assimilating snow depth observations is needed. For example, Fritzner et al. 2019 found assimilating snow depth observations had positive effects on short-term forecasts of snow depth and sea ice concentration.

This study uses different OSSEs to investigate how the non-Gaussian nature of different sea ice fields impacts data assimilation-generated sea ice analyses. Using OSSEs provides an experimental framework to test the impacts of synthetically generated observations in different data assimilation configurations. This study expands on previous research on sea ice data assimilation that was laid out by Zhang et al. (2018). The OSSEs presented in this study will test different experimental setups to investigate their impacts on sea ice and snow states generated by data assimilation. These experiments will investigate the impacts of post-processing updates for snow on top of sea ice, different assimilated observation combinations, and different data assimilation methods. This study highlights the impacts of the non-Gaussian nature of certain sea ice variables on the generation of sea ice analyses when using an EnKF data assimilation method. Section 2 describes the sea ice model and the data assimilation experimental setup along with the description of the different OSSEs that were completed. Section 3 presents the results obtained from the different OSSEs. Section 4 discusses the conclusions and future work on this research.

## 2   Methods and Experimental Setup

### 2.1   DART-CICE data assimilation system

For this study, the Los Alamos Sea Ice Model version 5 (CICE5; Hunke et al. 2015) is used to integrate the analyses forward in time while using an ensemble Kalman filter (EnKF) data assimilation technique to generate analyses. The Data Assimilation Research Testbed (DART; Anderson et al. 2009) software was used to implement the EnKF. Hereafter, we refer to this modeling configuration as CICE-DART. The CICE5 model setup closely follows that in Zhang et al. (2018) while the data assimilation setting will be different in the experiments.

### 2.1.1   DART

The data assimilation technique used in this study is the ensemble adjustment Kalman filter (EAKF; Anderson 2001), which is a modified version of the ensemble Kalman filter (Burgers et al., 1998) and a variation of the deterministic ensemble square-root filter (Tippett et al., 2003). The EAKF combines observations with an ensemble of short-term model forecasts over a specific observation window to produce an ensemble of the best estimate of the sea ice state. One important aspect of the

EAKF is its ability to use the ensemble to estimate a flow-dependent background-error covariance, which differs from a static background-error covariance typically employed by variational techniques. Additionally, a non-Gaussian rank histogram filter (RHF, filter option 8 in DART; Anderson 2010) is tested to compare with EAKF results. To reduce sampling errors due to limited ensemble member size, covariance localization was applied only in the horizontal direction. A Gaspri-Cohn fifth-order polynomial was applied in the horizontal directions to limit observation updates within a specified cutoff radius of 0.05 (i.e., ~320 km; Gaspari and Cohn 1999). Adaptive prior covariance inflation was applied by "inflating" the ensemble perturbations in prior background fields, increasing the variance by pushing ensemble members away from the ensemble mean (Anderson, 2007). Zhang et al. (2018) found a reduction in Arctic sea ice area and volume errors when prior inflation was applied in their study. Inflation damping is set to 0.9 to help control the growth of the inflation factor for the different state model variables. Any assimilated observation type is allowed to update all model state variables during the assimilation step unless otherwise noted. There was no cross-variable localization applied in this study.

### 2.1.2   CICE

CICE5 is the sea ice component within the Community Earth System Model (CESM; Danabasoglu et al. 2020) that is used to make climate projections. CICE5 simulates the evolution of sea ice and snow through the representation of thermodynamic and dynamical processes using an ice thickness distribution. The evolution of sea ice thickness, which is represented by the quotient of sea ice volume and sea ice area, is accomplished by partitioning the sea ice pack distribution within a grid cell into multiple thickness categories (Lipscomb, 2001). For this study, there are five thickness categories for both sea ice and snow with lower bounds of 0, 0.64, 1.39, 2.47, 4.57 m. Respecting the category bounds poses a challenge during the data assimilation step when updating sea ice area and sea ice volume. Snow depth is also partitioned into five categories. Each thickness category is divided into multiple layers (both sea ice and snow if present) to represent the evolution of sea ice temperature, salinity, and enthalpys related to sea ice and snow. CICE was coupled to a slab ocean model (SOM) that provides the ocean forcing in the form of annually periodic, prescribed ocean forcing data (e.g., sea surface temperatures, ocean heat fluxes). The atmospheric forcing data comes from the Community Atmosphere Model version 6 (CAM6)/Data Assimilation Research Testbed ensemble reanalysis (Raeder et al., 2021) for the time period of interest. Default namelist settings were used in this study (Hunke et al., 2015) except for perturbing several input CICE parameters, which will be discussed in the next section.

### 2.2   Perfect model OSSEs

Given the uncertainties and potential biases of satellite-retrieved sea ice and snow observations, this study applies perfect model OSSEs to investigate non-Gaussian impacts that could be introduced while assimilating these observations. Each ensemble consists of 80 CICE5 members because there are 80 different CAM6/DART reanalysis atmospheric forcing files. Each CICE5 ensemble member uses the same SOM forcing. To increase the ensemble spread, three different parameters were perturbed that impact albedo, heat transfer through snow, and the ability to move sea ice within the ocean. The standard deviation of the dry snow grain radius ($R_{snw}$) controls the optical properties of snow and is one of the key parameters that determines snow albedo in the solar radiation parameterization (Briegleb and Light, 2007). The thermal conductivity of snow ($k_{snw}$) directly impacts the

amount of heat that can be transferred through the snow pack, thereby affecting the evolution of sea ice (Sturm and Massom, 2017). The neutral ocean-ice drag coefficient (dragio) controls the horizontal momentum exchange at the ice-ocean interfaces, which determines the drag forces on the sea ice (Lu et al., 2011). These three parameters were chosen because they are among the top parameters that drive variability within CICE5 in both summer and winter (Urrego-Blanco et al., 2016). See the data availability statement for access to the perturbed parameter values used in this study. To achieve the climatological state of sea ice and snow, a single member is run for 40 years using periodic atmospheric forcing for the year 2012. To build our 80 member ensemble, we first used only 80 different atmospheric forcings to cycle over 2012 for 10 years to build in variability related to the atmosphere. Each ensemble member is then run for an additional 15 years, cycling over 2012, using the distinct atmospheric forcing and parameter set to generate free forecasts that can be used as a reference case (Fig. 1). One of the free forecast members is randomly chosen as the simulated "truth." For this study, the free forecast ensemble mean is negatively biased compared to the truth member for different sea ice and snow characteristics. The free forecasts will provide a reference for comparison with the different data assimilation experiments.

Since satellites can not retrieve multi-category model quantities, aggregate synthetic observations are generated from the truth member to produce sea ice concentration (SIC), sea ice thickness (SIT), snow depth ($D_{snow}$), and sea ice surface temperature (SIST). The multi-category state model variables that are updated via data assimilation or post-processing are sea ice area ($A_{ice,n}$), sea ice volume ($V_{ice,n}$), and snow volume ($V_{snow,n}$). When those multi-category state model variables are summed over the different categories, they are referred to as $A_{ice}$, $V_{ice}$, and $V_{snow}$. To compute data assimilation updates, it is necessary to compute an observation's expected value from the model state, which is called the forward operator. SIC is just the sum of the area values in the different thickness categories computed as

$$SIC = \sum_{n=1,5} A_{ice,n}. \tag{1}$$

The mean SIT of a grid cell is computed by summing the sea ice volumes in the different thickness categories and then dividing by the aggregated sea ice area as follows

$$SIT = \frac{\sum_{n=1,5} V_{ice,n}}{\sum_{n=1,5} A_{ice,n}}. \tag{2}$$

The mean $D_{snow}$ of a grid cell is computed in the same fashion as SIT except using summed snow volumes

$$D_{snow} = \frac{\sum_{n=1,5} V_{snow,n}}{\sum_{n=1,5} A_{ice,n}}. \tag{3}$$

The mean SIST of a grid cell is the area weighted mean temperature across the different thickness categories on the surface of the sea ice

$$SIST = \frac{\sum_{n=1,5} SIST_n * A_{ice,n}}{\sum_{n=1,5} A_{ice,n}} \tag{4}$$

where $SIST_n$ is the sea ice surface temperature for the different thickness categories. In this OSSE framework, synthetic observations are generated from the truth member using the forward operators and are assimilated. Normally, synthetic observations

are created by adding a draw from a normal distribution with a mean of zero and a specified observation error standard devia-
tion. This method was chosen to create the synthetic sea ice surface temperature observations that were assimilated. However,
sea ice and snow quantities have single (SIT, $D_{snow}$) and double (SIC) bounds in their representations. Because of this, we
will use a single (SIT,$D_{snow}$) and double (SIC) truncated normal distribution when generating the synthetic sea ice and snow
observations that are assimilated in our OSSEs. The observation error standard deviation for SIC is 15% of the true values
of SIC ($SIC_{error} = SIC_{truth}*0.15$; Zhang et al. 2018) and 0.1 m for SIT (approximation of future high precision data; Zhang
et al. 2018). While studies that use real SIT observations have varied their uncertainties depending on the thickness value (Xie
et al., 2018; Cheng et al., 2023), due to the complexity of computing SIT (Zygmuntowska et al., 2014) this study chose to
use a single value for SIT uncertainty. The SIT observation error of 0.1 m is a goal for future satellite platforms and is not
the observation error for current observing platforms. The observation error standard deviation is 10% of the true values of
$D_{snow}$ (approximation of future high precision data; Rostosky et al. 2020) and 1.5°C for SIST (Hall et al., 2015). Due to the
SIC observation error method, only synthetic SIC observations greater than 0.01 (approximately the precision found in passive
microwave sea ice concentration observation files, Meier et al. 2021) are assimilated. Similarly, the observation error for $D_{snow}$
has a lower bound of 0.005 m for synthetic observations close to zero. The locations for all synthetic observation types that are
assimilated were based on CryoSat-2 locations (locations measured every 10 seconds; more details on locations see CryoSat-
2 Product Handbook at https://earth.esa.int/eogateway/documents/20142/37627/CryoSat-Baseline-D-Product-Handbook.pdf),
which provides the observational network for testing (Fig. 2). While different sea ice observation networks in the real world
usually do not match, the observation network chosen for this study was chosen because of the easy experimental setup and
fair comparison between the synthetic observations that were assimilated in this study.

Six different experiments were completed to test different observation combinations, data assimilation techniques, and post-
processing updates (Table 1). EAKF-ConcThick is an extension of the work completed by Zhang et al. (2018) where they only
allowed observation increments to update the sea ice area in the different categories while updating the sea ice and snow volume
via post-processing. In EAKF-ConcThick, we allow the category-based sea ice area and volume to be updated independently
by synthetic SIC and SIT observations while updating snow volume via post-processing. The equations for post-processing
snow volume updates in the different categories are the following:

$$h_{snow,n}^{prior} = \frac{V_{snow,n}^{prior}}{A_{snow,n}^{prior}}, \tag{5}$$

$$V_{snow,n}^{posterior} = A_{ice,n}^{posterior} \times h_{snow,n}^{prior}, \tag{6}$$

where $A_{ice,n}^{prior}$ is the prior sea ice area in the different thickness categories,$V_{snow,n}^{prior}$ is the prior snow volume in the different
thickness categories, $A_{ice,n}^{posterior}$ is data assimilation updated sea ice area in the different categories and $h_{snow,n}^{prior}$ are the prior snow
thickness values in the different categories. In EAKF-ConcThickSnow, snow volume is no longer updated by post processing
and assimilation of synthetic $D_{snow}$ is included in the assimilated observation subset. Since real world snow depth observations
still have their limitations (Rostosky et al., 2018; Fritzner et al., 2019), the synthetic snow depth observations generated for this

OSSE will test the impacts if high-quality snow observations are available year-round in the future. All assimilated synethetic observations (SIC,SIT,SNWD) update all category-based model state variables ($A_{ice,n}, V_{ice,n}, V_{snow,n}$). To test the non-Gaussian effects of the synthetic SIC observations, EAKF-ThickSnow only assimilates synthetic SIT and $D_{snow}$ while allowing the category-based sea ice area, sea ice volume, and snow volume state variables to be updated from the observation increments. RHF-ConcThickSnow investigates the impacts of using a non-Gaussian data assimilation method, the rank histogram filter, when working with the non-Gaussian sea ice and snow variables in the CICE model. EAKF-ModifiedFO investigates the impacts of having sea ice thickness and snow depth output from CICE instead of having the forward operators within DART compute these quantities. This arises from the fact that prior inflation is applied, which can push either the sea ice area or sea ice volume below zero. Since computing sea ice thickness or snow depth is the division of either sea ice or snow volume by the sea ice area, this could lead to shuffling of the distribution if values become negative. Finally, EAKF-SIST tests the impacts of assimilating additional synthetic SIST observations to further improve the updates of sea ice and snow states. While synthetic SIST observations are assimilated, sea ice surface temperatures in the different thickness categories are not updated from the data assimilation step.

Due to the bounds related to sea ice and snow state variables, there are different conditions under which special treatment is needed to ensure that the respected bounds are met. SIC (summed sea ice area across the categories) must remain between 0 and 1. Similarly, sea ice and snow volumes (summed across the categories) must remain above zero. If negative values occur for SIC or the volumes, all categories are set to zero. Additionally, category-based sea ice area values are scaled if the SIC exceeds one after the assimilation updates. In the event SIC exceeds one, the scaling of the category-based sea ice area is as follow:

$$A_{ice,n} \; = \; A_{ice,n} * \frac{1}{SIC}. \tag{7}$$

In the case where SIC is within the bounds but individual categories become negative, those categories are set to zero and the remaining nonzero categories are reduced proportionally to compensate for the negative amount. Lastly, special care is taken to account for the cases where SIC is greater than zero but sea ice volume in all categories is zero. This can occur during data assimilation updates to the category-based sea ice volume (updates removes all the sea ice volume) or if the data assimilation updates create some amount of sea ice area but the sea ice volume remains zero. A new sea ice volume is computed by multiplying the average thickness value allowed in the associated category (0.32, 1.01, 1.93, 3.51, 6.95) by the sea ice area for the category.

The same initial conditions used to generate the free forecasts were used for the experiments listed in Table 1. The free forecasts provide a reference to the amount of variability that was generated during the spin-up process (Fig. 1). All experiments were initialized on 1 January 2013 and the cycling period was for the entire year 2013. In all experiments, observations were assimilated at a daily interval.

## 2.3 Model Verification Metrics

Time series of total sea ice area, sea ice volume, and snow volume will be ensemble mean forecast quantities used to evaluate
CICE-DART performance over the cycling period. The equations for computing total sea ice area and volume are as follows:

$$\text{Total-Sea-Ice-Area(t)} = \sum_{n=1,j} (\text{SIC(t)}_j * \text{grid-cell-area}_j), \tag{8}$$

$$\text{Total-Sea-Ice-Volume(t)} = \sum_{n=1,j} (\text{V}_{\text{ice}}(t)_j * \text{grid-cell-area}_j), \tag{9}$$

where t is time, j is the total number of grid points in the Northern Hemisphere and grid-cell-area is the area of the grid cell.
Total snow volume is computed in the same way as total sea ice volume but instead using snow volume. Spatial Probability
Score (SPS) is computed to investigate potential sea ice edge errors over the cycling period (Goessling and Jung, 2018).
Following Goessling and Jung 2018, the ice edge is defined using the 15% sea ice concentration contour in this study. Due to
data storage issues, SPS could not be calculated for EAKF-SIST. Additionally, ensemble mean spatial biases will be computed
for SIC, sea ice volume, and snow volume over different cycling periods. Welch's t-test will be applied to test for significant
biases (Welch, 1947). The ensemble mean was chosen because the statistics were nearly identical regardless of whether the
ensemble mean or ensemble median was used.

Mean absolute bias (MAB) and mean square error (MSE) will be computed over the time series of total sea ice area, sea ice
volume, and snow volume for additional performance evaluation. The equations for MAB and MSE are as follows:

$$\text{MAB} = \sum_{t=1}^{N} |X_i^m - X_i^t|, \tag{10}$$

$$\text{MSE} = \sum_{t=1}^{N} (X_i^m - X_i^t)^2, \tag{11}$$

where i is the time index, N is the total number of times (i.e, number of days), $X_i^m$ ensemble mean forecast quantity (e.g., total
sea ice area), and $X_i^m$ is the true value for the forecast quantity. The integrated ice-edge error (IIEE) is another forecast metric
that is applied to the ensemble mean, which is analogous to SPS when using a single, deterministic forecast (Goessling et al.,
2016). IIEE evaluates potential sea ice edge differences between the ensemble mean and the truth. IIEE is more suitable for user
forecast evaluation of the sea ice edge compared to the traditional sea ice extent (Tietsche et al., 2014). The IIEE is the sum of
the area grid boxes where the ensemble mean and the truth disagree on whether sea ice is present (over-prediction; $\text{SIC}_{\text{truth}}=0$
and $\text{SIC}_{\text{ensemble mean}}>0$) or not (under-prediction, $\text{SIC}_{\text{truth}}>0$ and $\text{SIC}_{\text{ensemble mean}}=0$). Similar to previous studies computing
IIEE, a SIC threshold of 15% is used to determine whether a grid cell is identified as having sea ice (Goessling and Jung,
2018; Zampieri et al., 2018). An attractive feature of IIEE is that it can be decomposed into an absolute extent error (AEE) and
a misplacement error (ME). AEE is the absolute difference (|over-prediction - under-prediction|) between predictions, which

can help determine whether there is a bias for over- or under-predicting sea ice coverage. MEE is the misplacement error (2 $\times$ **min**(over-prediction,under-prediction)) reflecting whether there is too much sea ice in one location and too little in another. IIEE along with its components AEE and ME will be computed daily. Welch's t-test was used to determine whether there were significant differences between MAB, MSE, and IIEE values between experiments. Finally, Spearman correlations are computed between the perturbed parameters and different CICE model outputs.

## 3 Results and discussion

### 3.1 Optimization of sea ice and snow data assimilation

The first three experiments investigate which assimilated synthetic observation subset produces the most accurate forecasts for both sea ice and snow. All the experiments have similar skill in predicting the sea ice edge and are better than the free forecast (Fig. 3A). However, there is a period during August and September when experiments assimilating SIC, EAKF-ConcThick and EAKF-ConcThickSnow, have smaller errors in predicting the sea ice edge. Daily biases of total sea ice area, sea ice volume, and snow volume are computed throughout the cycling period to compare the performance of the experiments with the truth and free forecasts (Fig. 3B,C,D). Compared with the free forecast, EAKF-ConcThick performs better for both total sea ice area and sea ice volume. However, total sea ice area and sea ice volume were negatively biased from the start of the melt season in May until the re-freeze in September. Total snow volume for EAKF-ConcThick is comparable to the free forecasts. This means that the post-processing updates for the snow state variable are not as accurate compared to the sea ice state variables, which are updated directly from the multivariate data assimilation step. For EAKF-ConcThickSnow, there is little impact on biases associated with sea ice quantities. The biases associated with total snow volume are reduced in the EAKF-ConcThickSnow compared to EAKF-ConcThick and the free forecasts. This highlights the potential impacts snow depth observations could have if assimilated year-round which due to limitations is not possible (Rostosky et al., 2018). The negative biases found for total sea ice and sea ice volume during the summer for the first two experiments are now near zero for EAKF-ThickSnow. Improvements in total snow volume for EAKF-ThickSnow are isolated to the start of the melt season, however, the biases are similar to the first two experiments after this period. Regardless of these improvements, total snow volume is negatively biased throughout the cycling period for experiments where $D_{snow}$ observations are assimilated. Additionally, the biases for total snow volume are larger during the winter seasons leading up to June and then approach zero thereafter for experiments where $D_{snow}$ observations are assimilated. This result could mean that it takes a seasonal cycle to pull ensemble snow values closer to the truth. Removing SIC observations from the assimilated observation subset eliminates an observation that is doubly-bounded and whose values approach both of the bounds. Since SIC observations are more likely to be affected by their associated bounds (bulk of SIC observations near 1 unless near marginal ice zone) this could be the driving factor for the poor forecasts in the first two experiments.

Temporal forecast metrics are computed over the cycling period to pin-point which experiment is more accurate (Fig. 4). EAKF-ConcThick and EAKF-ConcThickSnow have the lowest total IIEE and are significantly different from the free forecast and EAKF-ThickSnow. This means that both EAKF-ConcThick and EAKF-ConcThickSnow produce a more accurate forecast

of sea ice coverage over the cycling period. This might seem inconsistent since the EAKF-ThickSnow daily biases were smaller. EAKF-ThickSnow has sea ice area MSE and MAB that is lower and significantly different from the other experiments and the free forecast. This means that removing the SIC observations provided a more accurate forecast of the sea ice area, however, this did have a negative impact on predicting the sea ice edge in EAKF-ThickSnow. This indicates that SIC observations play an important role in maintaining the sea ice edge close to the truth. Additionally, all experiments performed better for sea ice volume compared with the free forecast, with EAKF-ThickSnow being the most accurate. For snow volume, EAKF-ConcThick is not statistically better than the free forecast, indicating that post-processing snow updates is not a favorable method. Once again, EAKF-ThickSnow performs the best for snow volume even though SIC observations are not assimilated. While not assimilating SIC observations improves most forecast metrics, these observations are crucial for accurately representing the sea ice edge.

While EAKF-ThickSnow provided the most accurate forecasts for aggregated quantities, such as total sea ice area, it is unclear where those improvements occurred spatially over the Arctic at the start of the melt season. To gain more insight into the improved results, May-through-June averaged spatial biases of SIC, $V_{ice}$ and $V_{snow}$ are computed for the free forecast and each of the first three experiments (Fig. 5). For SIC, there are significant biases for the free forecast where the SIC values are too large over the central Arctic and too small near the marginal ice zone. EAKF-ConcThick and EAKF-ConcThickSnow show predominantly significant negative biases over the sea ice for SIC, whereas EAKF-ThickSnow reduces the spatial biases to near zero. The negative SIC spatial bias over the central Arctic explains why the total sea ice area for EAKF-ConcThick and EAKF-ConcThickSnow performed poorly compared to EAKF-ThickSnow. However, there are areas of larger bias value near the marginal ice zone for EAKF-ThickSnow, meaning it was less accurate in representing the sea ice edge. While all experiments reduced the magnitude of the $V_{ice}$ spatial bias, there is still an overall significant negative bias for EAKF-ConcThick and EAKF-ConcThickSnow. The spatial biases for EAKF-ThickSnow are near zero, and there are essentially no areas of significant bias. For $V_{snow}$, there are differences between the spatial biases for EAKF-ConcThick and EAKF-ConcThickSnow, highlighting the benefits of assimilating $D_{snow}$ observation over post-processing $V_{snow}$ updates. In EAKF-ThickSnow, there is an overall reduction in the significant negative biases over the central Arctic compared with EAKF-ConcThickSnow. In EAKF-ConcThick and EAKF-ConcThickSnow, the SIC observations have a negative impact on both the observed and non-observed model state variables. Removing SIC observations from the assimilated observation subset reduced the spatial coverage of significant biases for all state model variables.

An analysis increment indicates how the observations are pushing or pulling state model variables. Evaluating analysis increments will help determine how the assimilation of synthetic SIC observations impact the different data assimilation experiments. For EAKF-ThickSnow, there is a reduction in the magnitude of the spatial analysis increments at the start of the melt season compared with EAKF-ConcThick and EAKF-ConcThickSnow (Fig. 6A). The analysis increment reduction is mainly located over the central part of the Arctic, where SIC values for all ensemble members are close to 1. This implies that the assimilation of SIC observations leads to low biased SIC analyses. The SIC analysis increments become more similar across the experiments as one moves away from the central Arctic toward the marginal ice zone. The analysis increment patterns and magnitudes near the marginal ice zone for EAKF-ConcThick are less different than one might expect because of the increase

in IIEE. However, these analysis increments are averaged from May through June; therefore, the IIEE might be picking up on sea ice edge errors at different times throughout the cycling period. This is similar for the $V_{ice}$, where there is a reduction in the analysis increment magnitude over the central Arctic for EAKF-ThickSnow compared with EAKF-ConcThick and EAKF-ConcThickSnow (Fig. 6B). For $V_{snow}$ analysis increments, there is a flip in the sign between EAKF-ConcThick and EAKF-ConcThickSnow (Fig. 6). The negative $V_{snow}$ analysis increments in EAKF-ConcThick are connected to the SIC analysis increments due to the equation for post-processing (Equations 5 and 6). Since SIC analysis increments are mainly negative over the central Arctic, this would also lead to negative $V_{snow}$ analysis increments over this region due to the post-processing method. The differences in $V_{snow}$ analysis increments between EAKF-ConcThickSnow and EAKF-ThickSnow are small, indicating that the removal of synthetic SIC observations from the assimilated subset does not have a negative impact on the adjustments. Overall, EAKF-ThickSnow provides the best setup for sea ice and snow data assimilation. Even with a slightly higher IIEE, the removal of the synthetic SIC observations from the assimilate observation subset did provide better results. Further investigation is needed to understand the reason behind the persistent negatively biased total snow volume compared to the truth.

### 3.2    Further discussion on sea ice data assimilation

The removal of SIC as an assimilated synthetic observation improved forecasts of total sea ice, however, forecasts of the sea ice edge were less accurate according to the total IIEE and SPS. This result indicates that near the marginal ice zone there are benefits to assimilating SIC observations and that SIT observations provide poor multivariate updates for $A_{ice,n}$. Three additional experiments were completed to investigate the impacts on sea ice when using a non-Gaussian RHF, modified forward operators for synthetic thickness observations, and the assimilation of synthetic SISTs. Each additional experiment is compared with EAKF-ThickSnow. Sea ice edge errors are lower in RHF-ConcThickSnow, and the errors are larger in EAKF-ModifiedFO compared with EAKF-ThickSnow (Fig. 7A). Once again, this result highlights improvements when assimilating SIC observations near the sea ice edge. RHF-ConcThickSNow performs worse than EAKF-ThickSnow during the summer according to daily biases of total sea ice area, sea ice volume, and snow volume (Fig. 7B,C,D). The use of the non-Gaussian RHF did not handle the SIC observations better. Compared to EAKF-ThickSnow, EAKF-ModifiedFO and EAKF-SIST have similar daily biases for total sea ice and snow volume, however, not for total sea ice area. EAKF-ModifiedFO and EAKF-SIST have persistent, larger daily biases during summer compared to EAKF-ThickSnow. There does appear to be a slight improvement in total snow volume for EAKF-SIST compared with EAKF-ThickSnow during May, however, there are still negative biases throughout the cycling period.

RHF-ConcThickSnow does the best job representing sea ice coverage since its total IIEE is the lowest and it is significantly different from the other experiments (Fig. 8). RHF-ConcThickSnow assimilates SIC observations, which is likely why it is similar to our previous result from EAKF-ConcThickSnow (compare Fig. 4A with Fig. 8A). EAKF-ModifiedFO and EAKF-SIST essentially have the same total IIEE, which is statistically worse than EAKF-ThickSnow. The modification of the forward operator along with assimilating SIST observations does not improve the representation of sea ice coverage. For total sea ice area and sea ice volume, RHF-ConcThickSnow has the largest aggregated errors that are significantly different from the

other experiments (Fig 8B, C). This result is similar to EAKF-ConcThickSnow, where SIC observations were assimilated. While EAKF-ThickSnow does the best job representing the total sea ice area and sea ice volume, one thing that needs to be mentioned is that EAKF-SIST uses a modified forward operator. Since the sea ice statistics appear very similar between EAKF-ModifiedFO and EAKF-SIST, the modified forward operator could explain why the results for EAKF-SIST are worse than those for EAKF-ThickSnow.

Evaluating SIC over the start of the melt season (May-through-June) reveals that RHF-ConcThickSnow mostly has significant negative biases compared to the truth (Fig. 9A). This result is similar to EAKF-ConcThickSnow, where the EAKF is used instead of the RHF. Compared to EAKF-ThickSnow, there are larger, positive SIC biases for EAKF-ModifiedFO and EAKF-SIST near the marginal ice zone. These biased areas are mainly located in the Baffin Bay, Greenland Sea, and Barents Sea. The poor representation of the marginal ice zone for EAKF-ModifiedFO and EAKF-SIST could explain the larger total IIEE compared to EAKF-ThickSnow. RHF-ConcThickSnow has significant negative sea ice volume biases over most of the sea ice pack (Fig. 9B). Again, this agrees with the spatial biases for EAKF-ConcThickSnow over this period, further showing that switching to the RHF over the EAKF did not help alleviate the impacts of the SIC observations. The spatial biases of sea ice volume for EAKF-ModifiedFO and EAKF-SIST closely resemble those found in EAKF-ThickSnow, except near the marginal ice zone. The modified forward operator might introduce poor marginal ice zone updates without the constraint of SIC observations in this region. Overall, switching the data assimilation filter type did not resolve the issues related to assimilating SIC observations, and there are potential issues with using the modified forward operator near the marginal ice zone. However, the spatial biases are similar over most of the central Arctic, indicating that further investigation is needed to determine the negative impacts of the modified forward operator.

### 3.3 Simplified Data Assimilation Experiment

To further investigate the poor results obtained when assimilating SIC observations, a simplified data assimilation experiment was setup. This simplified DA experiment mimics SIC during wintertime over the pole, meaning that the true SIC does not change over time. With a constant truth value that does not change, synthetic observations are created that will be assimilated over the cycling period. The true SIC value is set to 0.99, and its corresponding observation error will vary from 0.1485 (value if using the same method as the OSSE experiments), 0.07425, and 0.037125. Two different filters, EAKF and RHF, will be tested using different observation error specifications. The initial ensemble spread has a standard deviation of 0.0142. No prior inflation was applied in these experiments. Six mini experiments were completed using a combination of different filter types (EAKF or RHF) and different specified observation errors. The experiments were cycled 5,000 times, assimilating the synthetic observations generated from the truth using a truncated normal distribution. These experiments will work with SIC directly, meaning that there are no thickness categories as in CICE. This means that the mapping between observation space to state space is linear, further simplifying this data assimilation experiment.

For all experiments, the prior ensemble mean drifts away from the true value and moves toward the average observation value over the cycling period (Fig. 10). The average observation value depends on the observation error specification; a smaller error leads to an average observation closer to the truth. For the largest observation error, the EAKF drifts toward the average

observation value at a quicker rate compared with the RHF (Fig. 10A,D). The slower rate exhibited by the RHF could mean that the filter weights the observations less compared with the EAKF. Even as the observation errors decrease, both the EAKF and the RHF move away from the truth and drift toward the average observation value (Fig. 10B,C,E,F). These experiments highlight that a reduction in the observation error still results in the prior ensemble being negatively biased when using distributions and observations near a bound.

The fact that the prior ensemble mean moves away from the true value regardless of filter type and observation error value demonstrates that our data assimilation solution is biased. This is because our observation error distribution is a truncated normal, whereas the observation likelihood for EAKF and RHF is assumed to be normal. Applying a non-Gaussian distribution for observation errors while using a Gaussian observation likelihood can lead to erroneous observation impacts, biasing analysis estimates (Pires et al., 2010; Fowler and Jan Van Leeuwen, 2013). This negative bias is exacerbated by the effects of prior inflation in our OSSEs by increasing prior variance, which further weights the observations even more. A better choice might be a combination of distributions representing the prior state and the observation errors more appropriately, as laid out in Anderson (2022).

### 3.4 Further discussion on snow data assimilation

Regardless of the first three experiments, the daily biases for snow volume are negative throughout much of the entire cycling period compared to the truth (Figs. 3D). Even the daily biases for the additional experiments are mainly negative throughout the cycling period (Fig. 7D). Further investigation is needed to fully understand why the snow volume is negatively biased regardless of the experimental setup. EAKF-ThickSnow will be further evaluated to investigate the reason for the low bias in snow volume. Since the ocean forcing is the same across ensemble members, the atmospheric forcing is evaluated for the ensemble mean. Breaking down the individual atmospheric heat fluxes, the shortwave radiation has the largest bias compared to the truth (Fig. 11A). The other atmospheric heat fluxes have smaller and near zero biases for most of the cycling period. The positive shortwave heat flux bias occurs during sunrise over the Arctic, which also corresponds to the period in EAKF-ThickSnow where the daily biases for snow volume are the largest (Fig. 3C). The spread in the absorbed shortwave heat flux grows during the onset into summer, which is during the start of the snow melt season (Fig. 11B). On average, the ensemble has absorbed too much incoming shortwave radiation compared to the truth. Interestingly, the spread of the absorbed shortwave heat flux collapses at the start of July, when the snow on top of the sea ice is at its minimum (Fig. 3C). One feature that can impact the absorbed shortwave radiation and is connected to snow cover is surface albedo (Fig. 11C). During the same period, the spread in the absorbed shortwave heat flux increases and the spread in the surface albedo spread increases. The spread in the surface albedo then collapses, similar to the incoming shortwave radiation heat flux, near the beginning of July. The surface albedo is, on average, too small compared to the truth, which could be the reason for the positive bias in the absorbed shortwave radiation. Lastly, the ensemble members almost appear to be sorted for both absorbed shortwave radiation and mean surface albedo, hinting that something systematic drives these quantities.

One potential reason that could be driving the negative biases found for the ensemble mean snow volume is that the snowfall originating from the atmospheric forcing file for the truth member is an outlier. This does not appear to be the case when

comparing daily biases of snowfall with the ensemble mean (Fig. 11D). The snowfall biases for the ensemble mean are near zero and fluctuate about the zero line, indicating that there is no clear systematic difference from the truth. One issue that has not been discussed is the role that the CICE perturbed parameters could play in snow evolution. Perturbed parameters have been used over the years to create more spread in atmospheric models (Murphy et al., 2004; Stainforth et al., 2005; Christensen et al., 2015; Orth et al., 2016), where the system is more chaotic. However, the impact the perturbed parameters would have on a less-chaotic system such as the cryosphere is unclear. Concerning total snow volume, there are larger and more significant correlations between the $R_{snw}$ parameter compared with the other perturbed parameters throughout the cycling period (Fig. 12A). The positive correlations indicate that larger standard deviations of the dry snow grain radius lead to greater total snow volume. This connection is a result of the larger standard deviations of dry snow grain radius resulting in a higher albedo, reflecting more incoming shortwave radiation (Hunke et al., 2015). Looking at snow melt, there are negative and significant correlations during the melt season for the $R_{snw}$ parameter, while the other parameters have little significant correlations (Fig. 12B). This means there is more snow melt for lower standard deviations of dry snow grain radius, resulting in more absorbed shortwave radiation due to a lower surface albedo. The $R_{snw}$ parameter for the truth member is located above the 75th percentile compared with the rest of the perturbed $R_{snw}$ parameters (Fig. 12C). Even with snow assimilation updates, the impact of the perturbed $R_{snw}$ parameter might play a larger role in snow evolution. Due to this fact, it is not surprising to find that the ensemble mean is negatively biased compared to the truth for total snow volume.

## 4    Conclusions

To advance our understanding of the global climate, it is critical to improve our representation of the different underlying Earth-system components within our coupled numerical climate models. One important Earth-system component–the cryosphere–has gathered recent attention due to declining Arctic summer sea ice and its link back to Arctic amplification. Data assimilation methods, such as the ensemble Kalman filter (EnKF), are one way to improve the representation of sea ice states by exploiting information from observations taken from satellites. However, the formulation of the EnKF has Gaussian assumptions and most state variables representing sea ice have some form of boundedness, which can lead to non-Gaussian distributions near those bounds. This study investigates the data assimilation impacts of the non-Gaussian nature of sea ice and snow variables on the generation of analyses within different observing system simulation experiments (OSSEs). The different OSSEs presented in this study will investigate which data assimilation setup provides the most accurate representation of sea ice and snow when dealing with non-Gaussian observations and state variables.

In this study, a sea ice model called CICE is coupled to the ensemble data assimilation software provided by DART to obtain a sea ice modeling system called CICE-DART. CICE-DART is used to conduct OSSEs to test different data assimilation configurations and the assimilation of different sea ice and snow observation subsets synthetically generated from a truth member. Six different experiments were completed to test different observation combinations, data assimilation techniques, and post-processing updates (Table 1).

The first three experiments explore the impact of different assimilated synthetic observation subsets on the generation of the most accurate forecasts for both sea ice and snow states. According to the daily biases and aggregated statistics, EAKF-ThickSnow is more accurate, when compared to the truth, for sea ice area, sea ice volume, and snow volume. This highlights the negative impacts that SIC observations have on forecasts when they are assimilated in EAKF-ConcThick and EAKF-ConcThickSnow. This result contradicts previous studies that found positive impacts from assimilating SIC observations (Sakov et al., 2012a; Massonnet et al., 2015; Posey et al., 2015). However, this result could be linked to differences in the observation error specification chosen for SIC observations in the different studies. In our study, early springtime SIC truth values are still close to one, maximizing their observation error (15% of the truth value), which leads to synthetic SIC observations being drawn further below the truth due to the bound at one. In addition, the prior spread increases because of the onset of springtime melt and prior inflation. Combining the low-bias observations with the increase in the prior spread leads to an enhancement of the non-Gaussian effects during early springtime. A similar but opposite effect (high-biased SIC observations) would be observed during winter; however, prior ensemble spread in the modeled SIC fields is smaller, resulting in a lower weighting of SIC observations. While potentially different from other studies, our chosen SIC observation error specification intensified the non-Gaussian effects of assimilating SIC observations while also showing the potential impact accurate SIT observations can have during data assimilation multivariate updating. Interestingly, SIC observations do provide positive updates in the marginal ice zone, as shown by SPS and total IIEE being lower in EAKF-ConcThick and EAKF-ConcThickSnow. Because of positive updates in the marginal ice zone, it would be optimal to assimilate SIC observations within the data assimilation system.

Additional OSSEs are performed to further investigate potential data assimilation improvements for sea ice (Table 1). A non-Gaussian RHF was tested since it was developed for non-Gaussian situations. The results showed little improvement over the EAKF when assimilating SIC observations. This is likely linked back to the RHF making some non-Gaussian assumptions on the tails and assumed normal likelihood when the distribution is not bounded. The modification to the forward operators did not improve sea ice data assimilation, especially regarding sea ice edge errors. This could mean that there are few instances of shuffling the sea ice thickness distribution due to prior inflation. Additionally, the multivariate update between sea ice thickness observations and sea ice area might be the reason for the increase in sea ice edge errors. Lastly, assimilating SISTs did not lead to increased skill for sea ice variables. The correction between the SISTs and sea ice model variables might not be significant, leading to little improvement.

To better understand the assimilation impacts due to the SIC observations, a simplified data assimilation experiment is completed. This simplified experiment mimics central Arctic SIC during the winter time, meaning the truth does not change. Regardless of the filter type or observation error value, the prior ensemble mean moves away from the truth and closer to the average observation value during the cycling period. These experiments verified that near a bound, the performance of the EAKF and the RHF is suboptimal. We believe the suboptimal performance is linked back to using a truncated normal distribution as the observation error distribution while the observation likelihood for the EAKF and RHF is assumed to be normal. Future projects focusing on sea ice data assimilation might want to consider a different choice for the observation likelihood specification, similar to those laid out in Anderson (2022). This would include using distributions for the prior PDF

and the observation likelihood that are similar to the observation error distribution and consider the bounds more appropriately (e.g., a truncated Gaussian distribution).

The evaluation of the additional OSSEs is performed to investigate their impact on snow updates. The improvements associated with using the non-Gaussian RHF over the EAKF were small for snow volume. This means that the non-Gaussian impacts from the SIC observations were negative for snow volume updates. Additionally, the modified forward operators have little impact on snow volume updates. However, there is a slight improvement in the snow volume when SISTs are assimilated. This improvement occurred during May and not over a specific area of sea ice. This could mean that the connections between SISTs and snow are more significant than those between sea ice, where the impacts were less impactful. Regardless, all additional experiments still experienced a negative bias throughout the entire cycling period. Further investigation revealed that one of the perturbed parameters could be driving the negative bias for snow volume. Correlations were larger and significant between snow variables and the representation of the dry snow grain radius size ($R_{snw}$) within our ensemble. Due to the random choice of the $R_{snw}$ parameter for the truth member, it is likely that the ensemble mean is negatively biased for snow volume.

Future work will further investigate how to properly assimilate SIC observations. Because of their positive impact on the marginal ice zone, an experiment could be proposed in which only SIC observations are assimilated in that remote location. Additionally, further investigation is needed to test the use of more sophisticated data assimilation methods that accurately handle non-Gaussian distributions. The RHF can represent non-Gaussian priors and arbitrary likelihoods for the observed variables. The RHF can be modified to work with bounded quantities (Anderson, 2020, 2022), which should be investigated in future studies. Lastly, supplementary OSSE experiments could be completed with a different ensemble member chosen as the truth to further understand the impacts of the perturbed parameters on representing snow volume. These additional experiments will further help us understand the correct data assimilation setup for representing sea ice and snow in climate analyses.

*Code availability.* CICE version 5 used for the experiments described here is part of the CESM2 framework which is publicly available for download from https://www.cesm.ucar.edu/models/cesm2/download.html (accessed in August of 2020). The data assimilation software used here can be downloaded from https://github.com/NCAR/DART (accessed in August of 2020).

*Data availability.* The perturbed CICE parameters used in this study is publicly available for download from https://doi.org/10.5281/zenodo.8164431. Post-processed and raw data from the experiments described here are stored on NCAR's campaign storage.

*Author contributions.* The development of the CICE-DART framework was completed by both CR and JA. CR prepared and performed the experiments under the supervision of JA. The manuscript was composed by CR with contributions and feedback from JA.

*Competing interests.* The contact author has declared that neither they nor their co-author have any competing interests.

*Acknowledgements.* This material is based upon work supported by the National Center for Atmospheric Research, which is a major facility sponsored by the National Science Foundation under Cooperative Agreement No. 1755088. Special thanks to the entire DART team for providing helpful input and source code support. I would also like to thank Cecilia Bitz and Molly Wieringa for their fruitful discussions. We thank the editor and anonymous reviewers for constructive comments that helped improve the manuscript.

515

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

**Table 1.** List of CICE-DART OSSEs with the different configurations.

| Experiments | Assimilated Observations | Modified Forward Operator | Postprocessed States | Assimilation Algorithm | Data Assimilation Updated State Vector |
|---|---|---|---|---|---|
| EAKF-ConcThick | SIC<br>SIT | No | Category<br>Snow Volume | EAKF | Category Sea Ice Area<br>Category Sea Ice Volume |
| EAKF-ConcThickSnow | SIC<br>SIT<br>$D_{snow}$ | No | No | EAKF | Category Sea Ice Area<br>Category Sea Ice Volume<br>Category Snow Volume |
| EAKF-ThickSnow | SIT<br>$D_{snow}$ | No | No | EAKF | Category Sea Ice Area<br>Category Sea Ice Volume<br>Category Snow Volume |
| RHF-ConcThickSnow | SIC<br>SIT<br>$D_{snow}$ | No | No | RHF | Category Sea Ice Area<br>Category Sea Ice Volume<br>Category Snow Volume |
| EAKF-ModifiedFO | SIT<br>$D_{snow}$ | Yes | No | EAKF | Category Sea Ice Area<br>Category Sea Ice Volume<br>Category Snow Volume |
| EAKF-SIST | SIT<br>$D_{snow}$<br>SIST | No | No | EAKF | Category Sea Ice Area<br>Category Sea Ice Volume<br>Category Snow Volume |

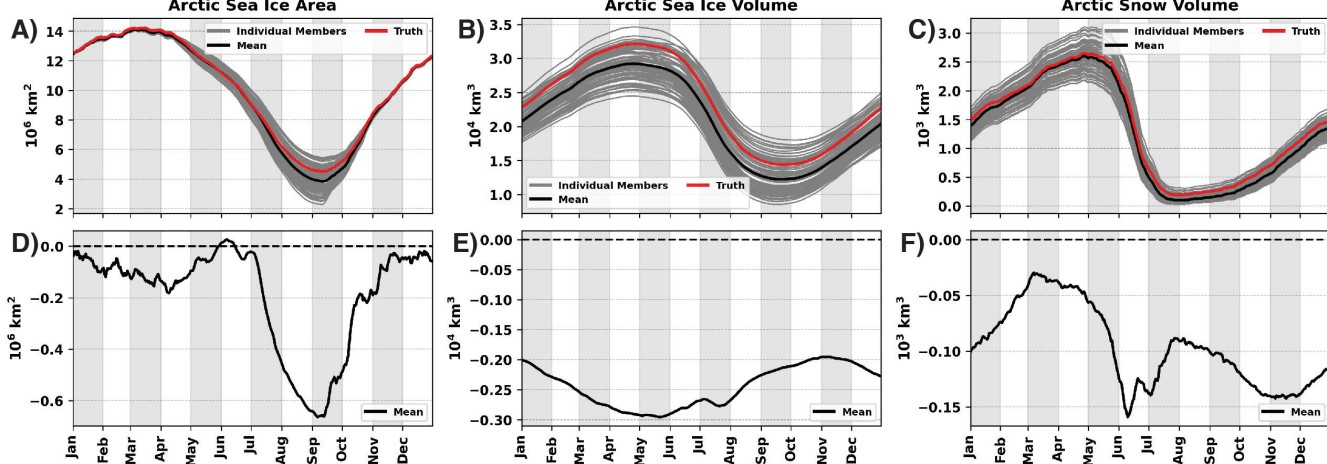

**Figure 1.** Daily total Arctic (A) sea ice area, (B) sea ice volume, and (C) snow volume from CICE5 free forecast simulations. Each gray line represents an individual ensemble member, black line represents the ensemble mean, and the red line represents the truth member. The truth member is a randomly selected ensemble member. Daily biases of the total Arctic (A) sea ice area, (B) sea ice volume, and (C) snow volume where the black line represents the ensemble mean difference compared to the truth. The black dashed line is the zero reference line. The free forecast period is for the year 2013.

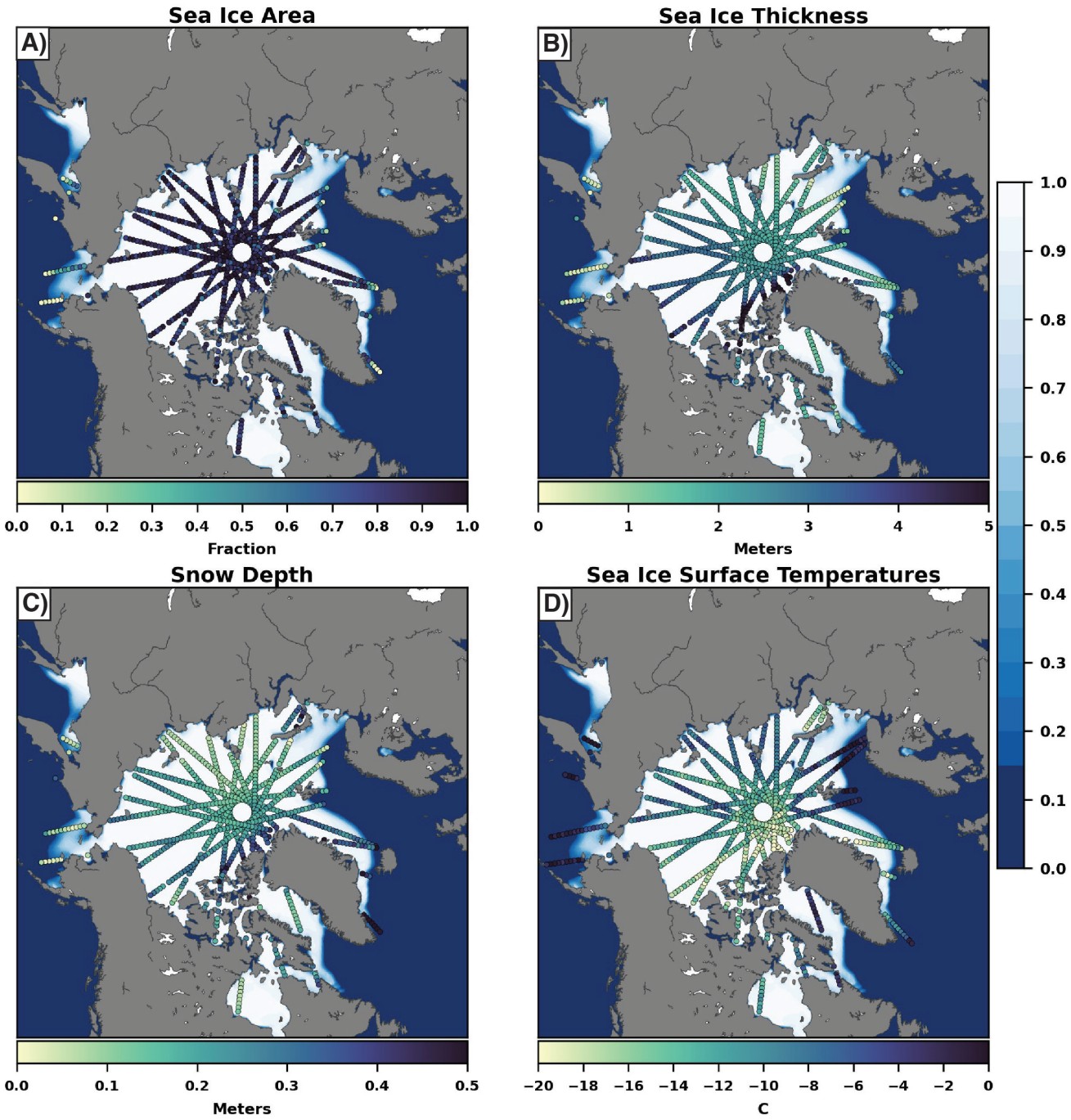

**Figure 2.** A snapshot example of the spatial locations of the OSSE synthetically generated (A) sea ice area, (B) sea ice thickness, (C) snow depth and (D) sea ice surface temperature observations that are assimilated. The observation locations are from Cryosat-2 latitude and longitude ground tracks. Colorfill is the ensemble mean of the sea ice area and the dots are the observation locations along with their associated value.

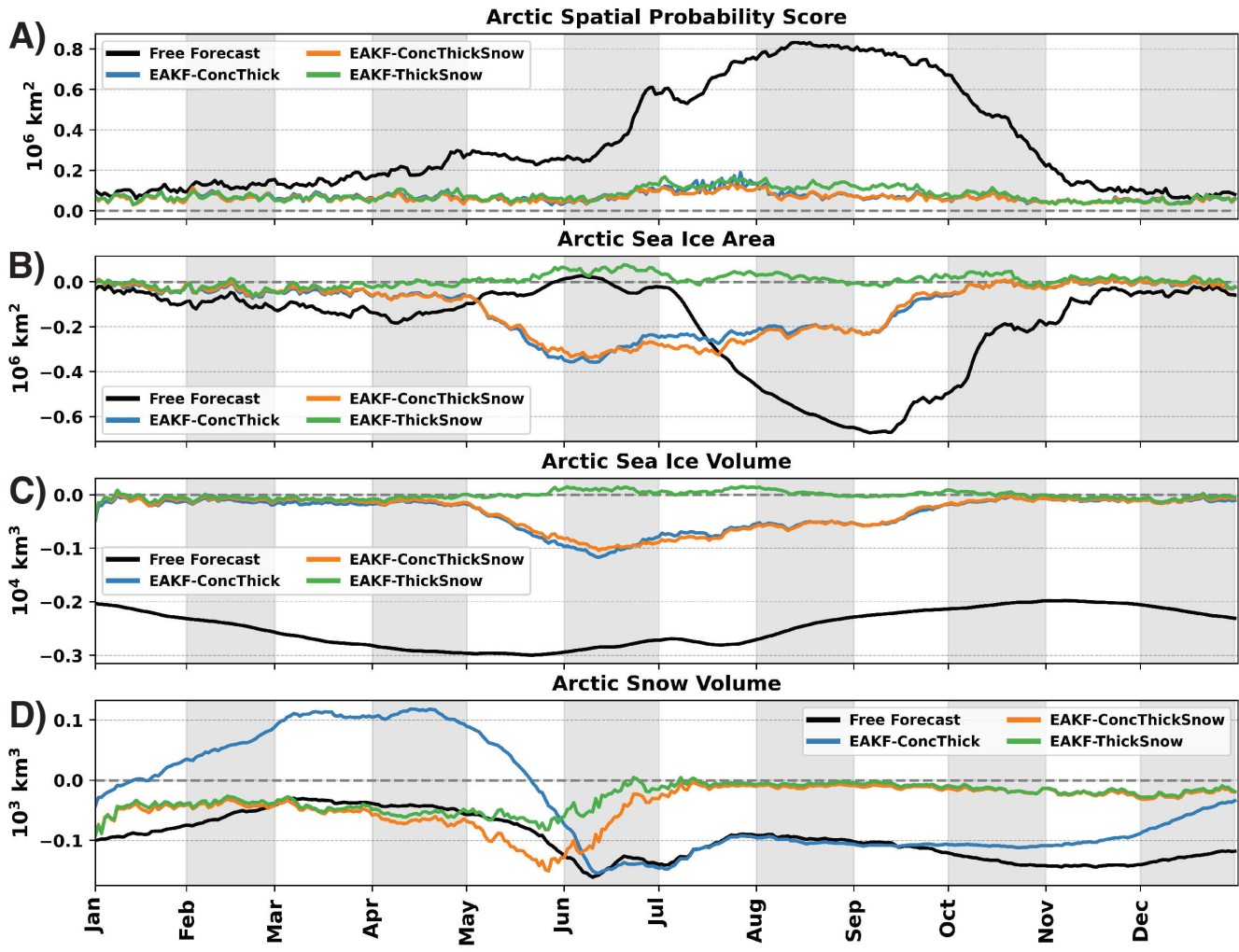

**Figure 3.** (A) Daily Arctic spatial probability score for the free forecasts, EAKF-ConcThick, EAKF-ConcThickSnow, and EAKF-ThickSnow. Daily biases of the Arctic total (B) sea ice area, (C) sea ice volume, and (D) snow volume from the free forecasts, EAKF-ConcThick, EAKF-ConcThickSnow, and EAKF-ThickSnow. Gray dashed lines are the zero reference line.

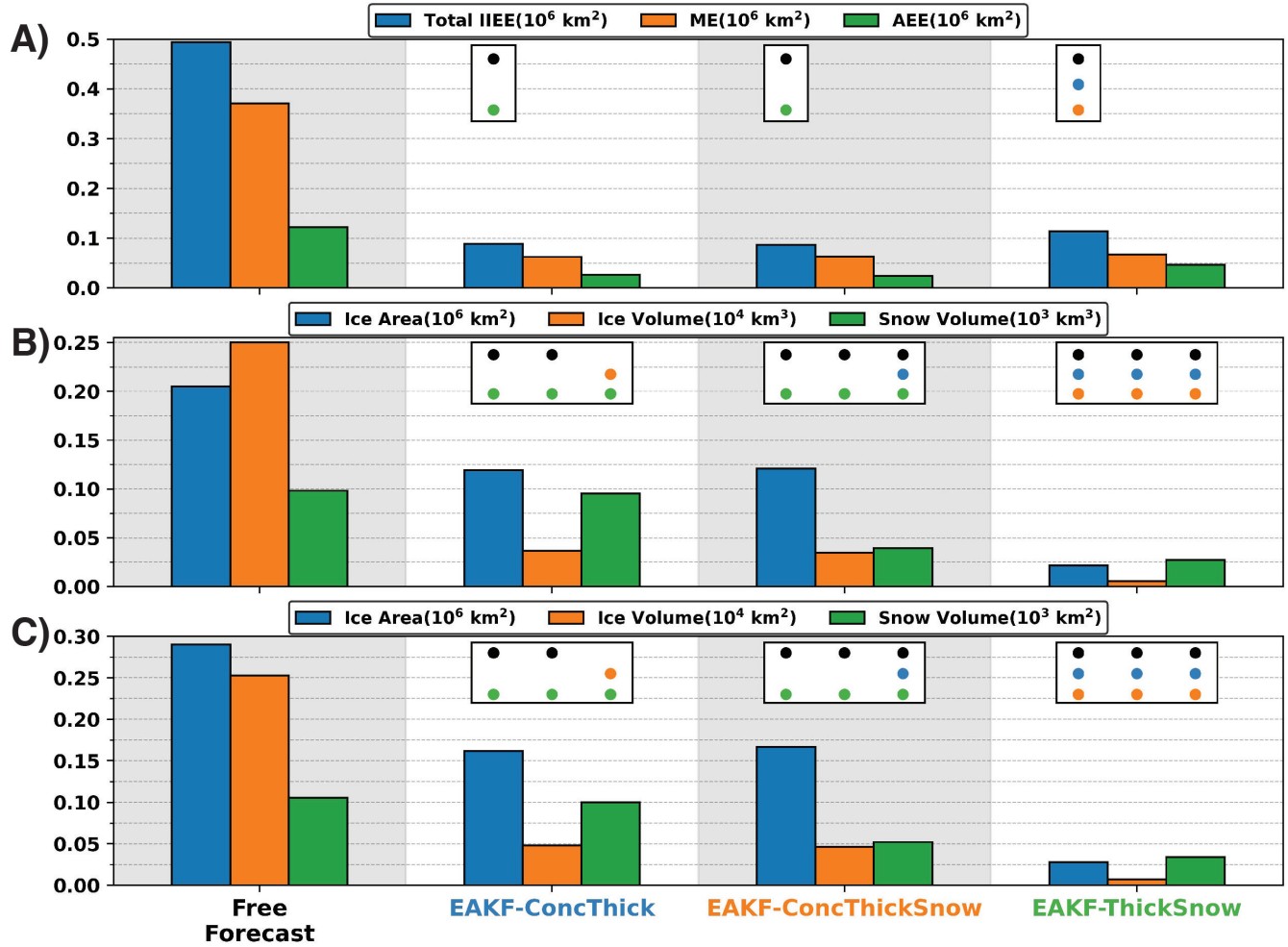

**Figure 4.** The (A) IIEE, (B) MAB, and (C) RMSE of sea ice area, sea ice volume and snow volume from the free forecast, EAKF-ConcThick, EAKF-ConcThickSnow, and EAKF-ThickSnow. Each index is computed using the ensemble mean and over the entire cycling period. Dots represents any pairs of experiments that are significantly different from a different experiment using a student t-test. Dot colors correspond to the different experiments.

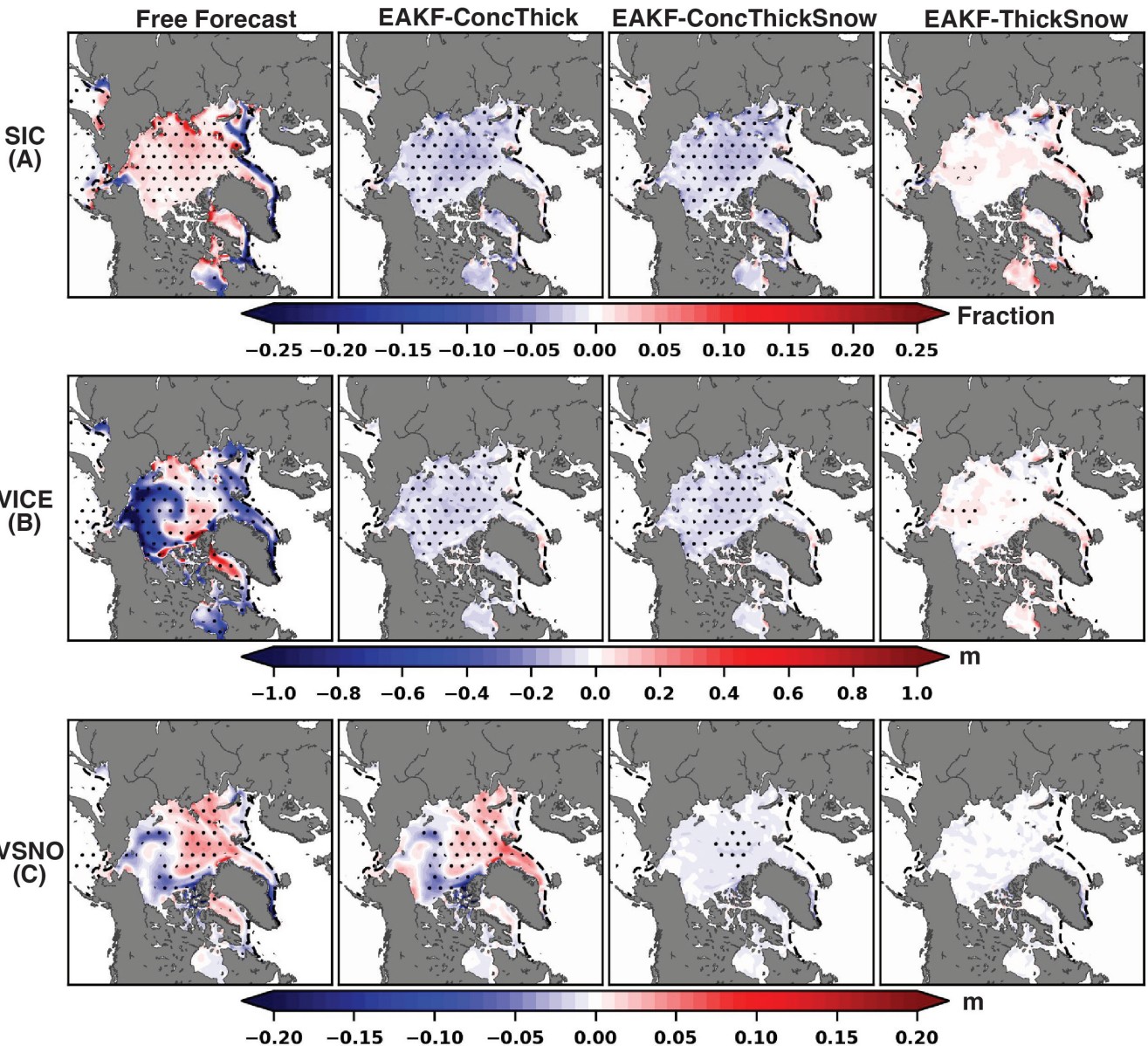

**Figure 5.** Ensemble mean spatial biases of (A) SIC, (B) $V_{ice}$, and (C) $V_{snow}$ averaged over May–June for the free forecast, EAKF-ConcThick, EAKF-ConcThickSnow, and EAKF-ThickSnow. Stippling represents significant biases at the 95% confidence interval using a Welch's t-test. The black dashed line is the sea ice edge (0.15 SIC).

.

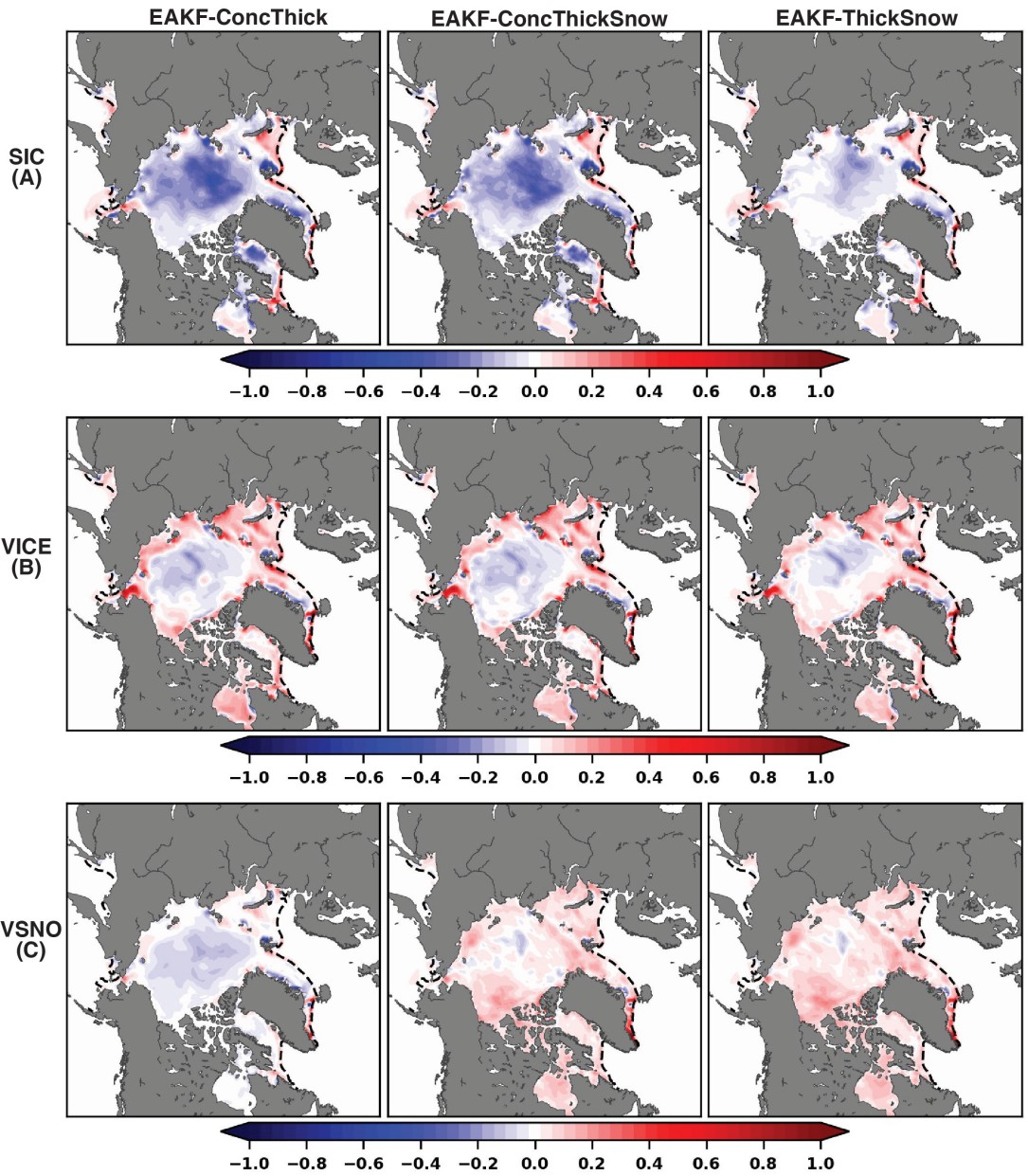

**Figure 6.** Normalized spatial analysis increments of (A) SIC, (B) $V_{ice}$, and (C) $V_{snow}$ averaged over May–June for EAKF-ConcThick, EAKF-ConcThickSnow, and EAKF-ThickSnow. Analysis increments of SIC, $V_{ice}$, and $V_{snow}$ were normalized using the largest absolute value from across the three experiments. The black dashed line is the sea ice edge (0.15 SIC).

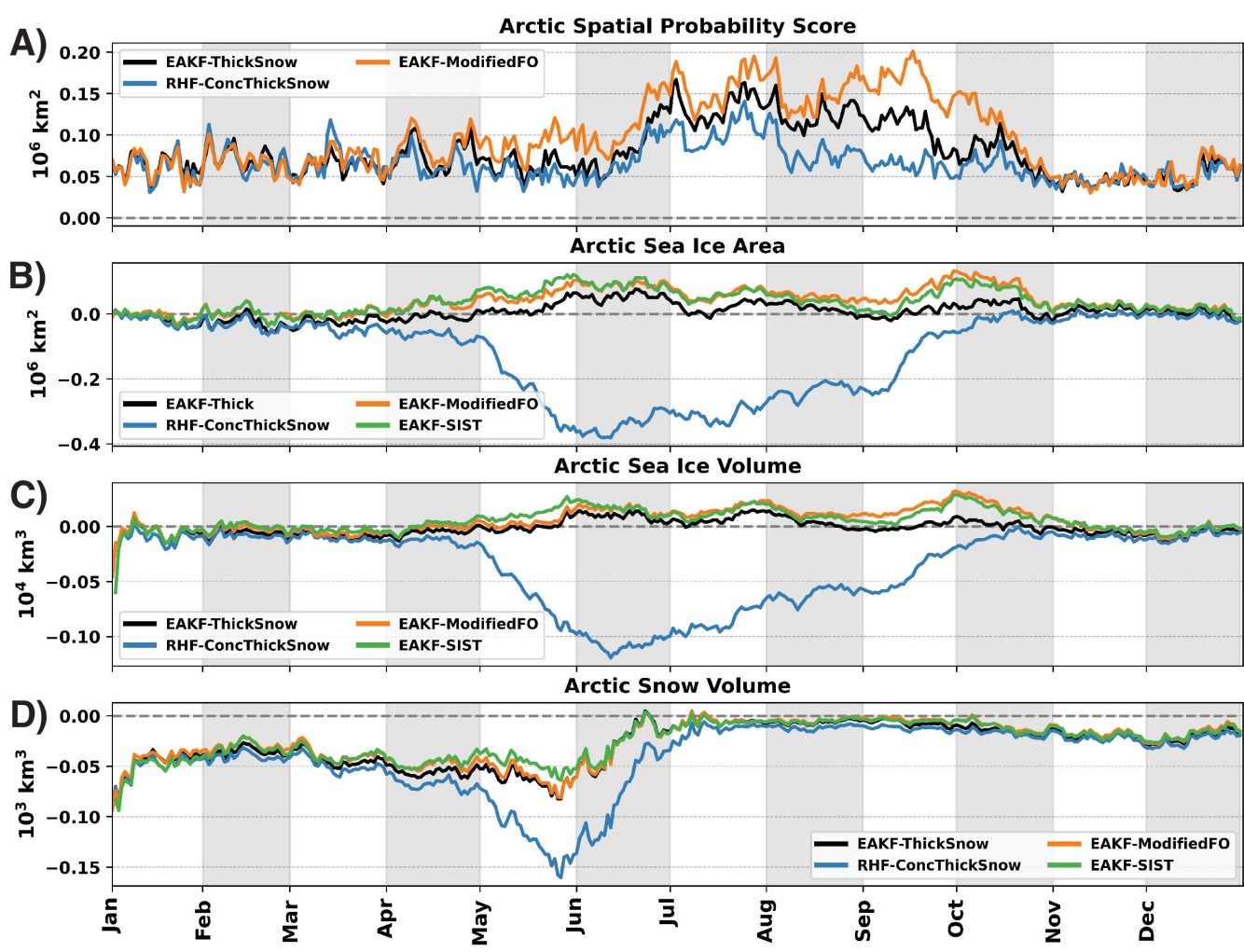

**Figure 7.** Same as Figure 3 but for EAKF-ThickSnow, RHF-ConcThickSnow, EAKF-ModifiedFO and EAKF-SIST.

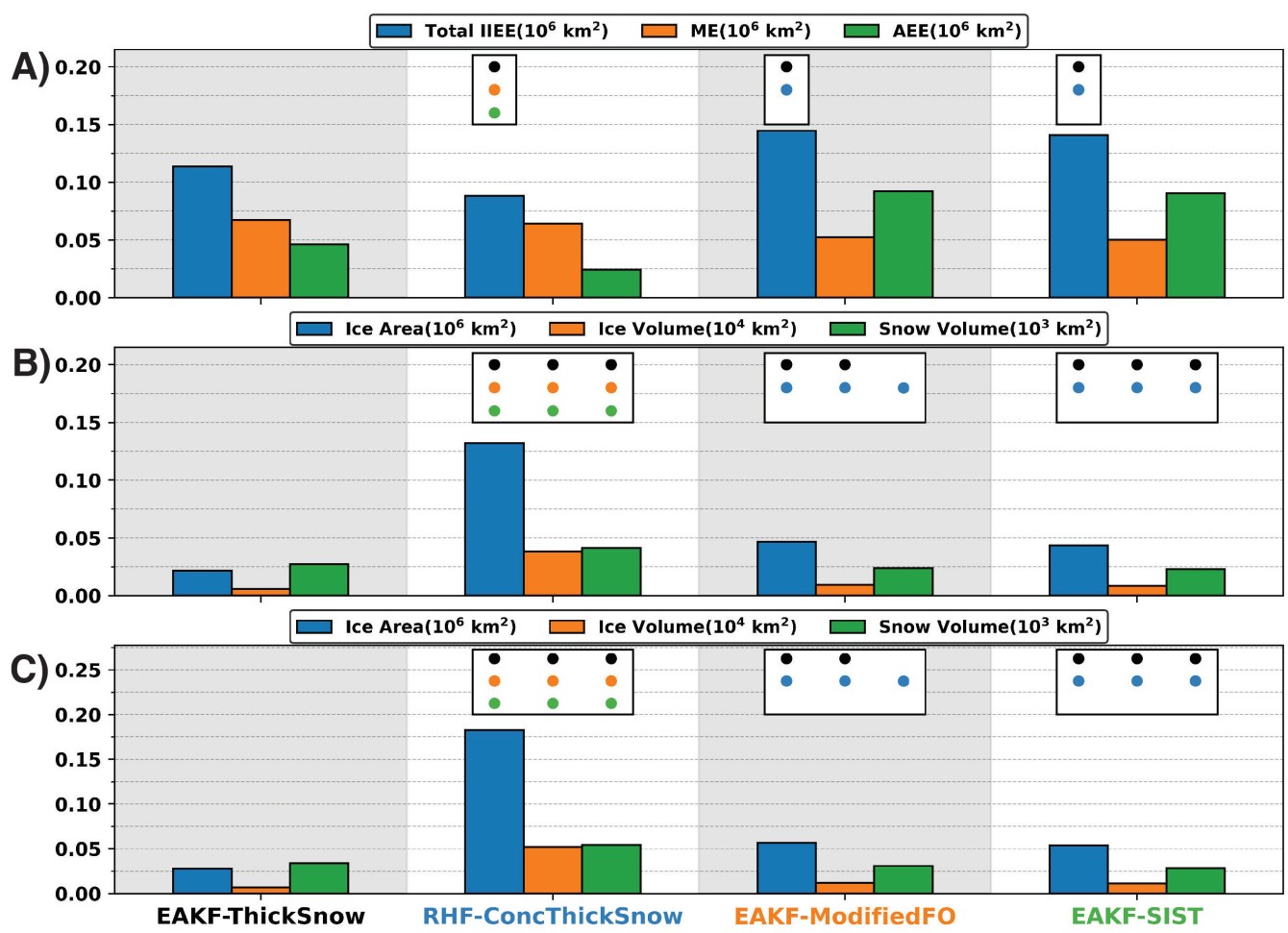

**Figure 8.** Same as Figure 4 but for EAKF-ThickSnow, RHF-ConcThickSnow, EAKF-ModifiedFO and EAKF-SIST

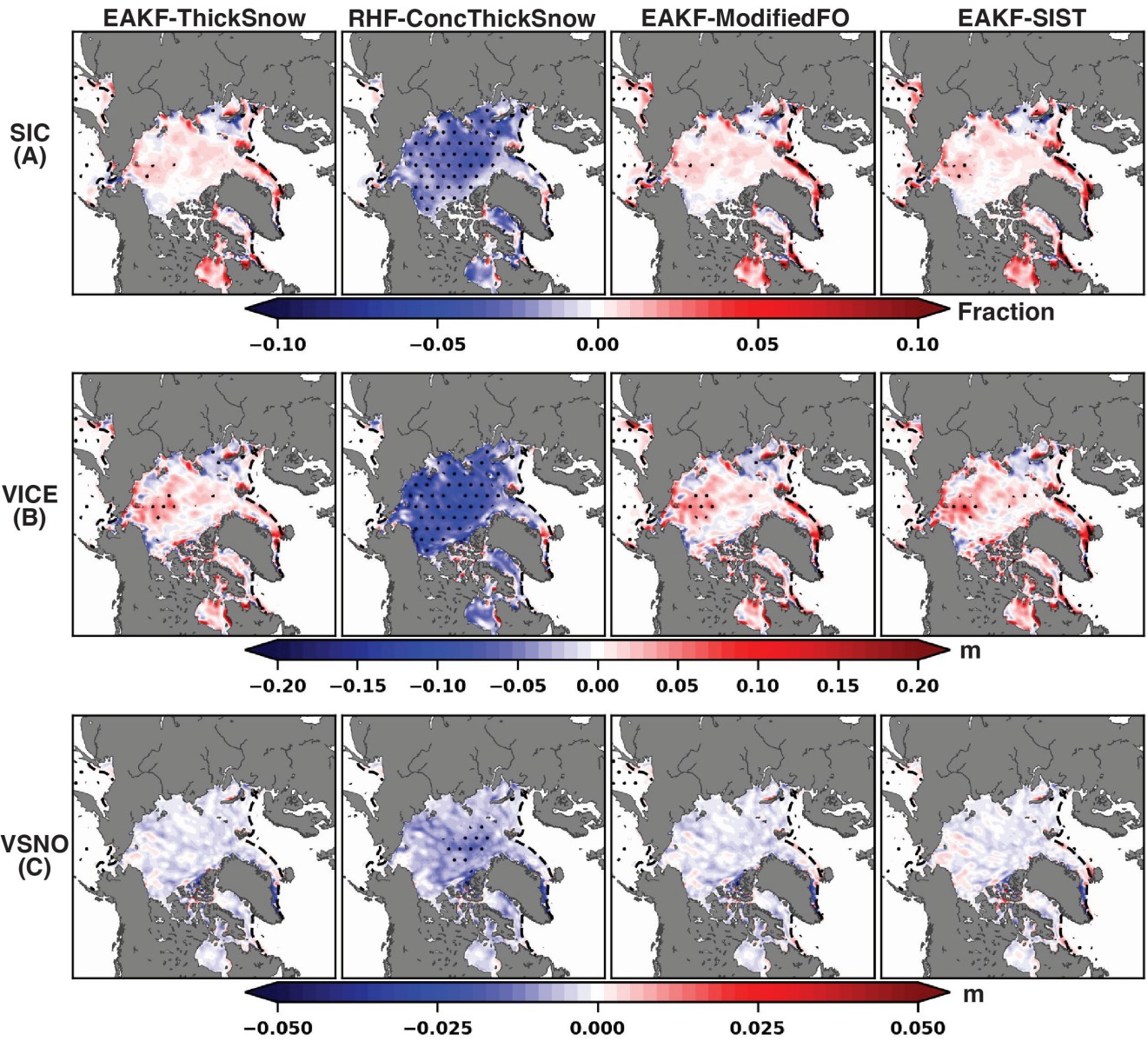

**Figure 9.** Same as Figure 5 but for EAKF-ThickSnow, RHF-ConcThickSnow, EAKF-ModifiedFO and EAKF-SIST.

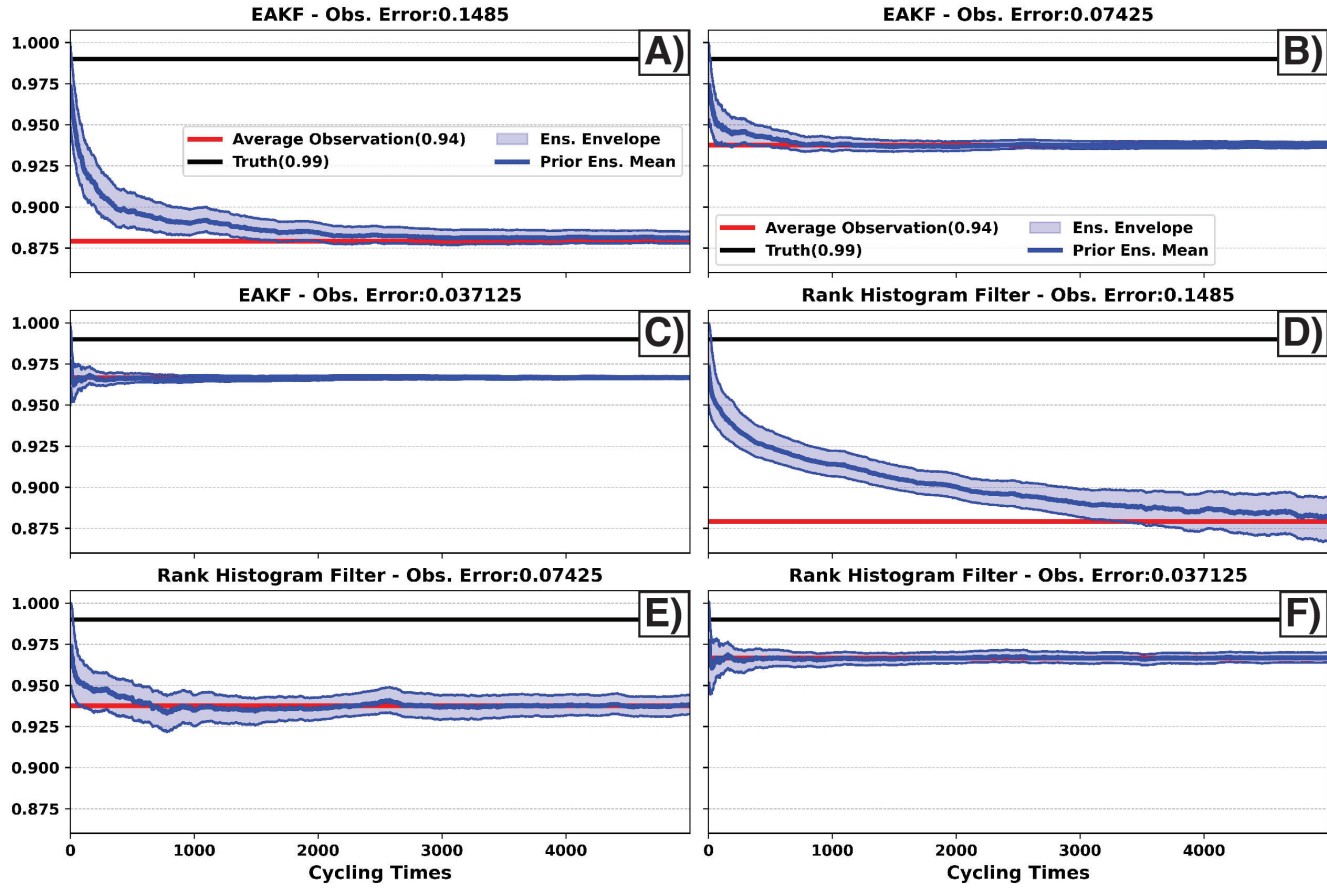

**Figure 10.** Prior ensemble mean (blue line) time series of SIC for experiments using (A,B,C) EAKF and (D,E,F) RHF. Each experiment was completed with observation error set to (A,D) 0.1485, (B,E) 0.07425, and (C,F) 0.037125. The red line represents the average observation value over the cycling period. The black line represents the true value over the cycling period.

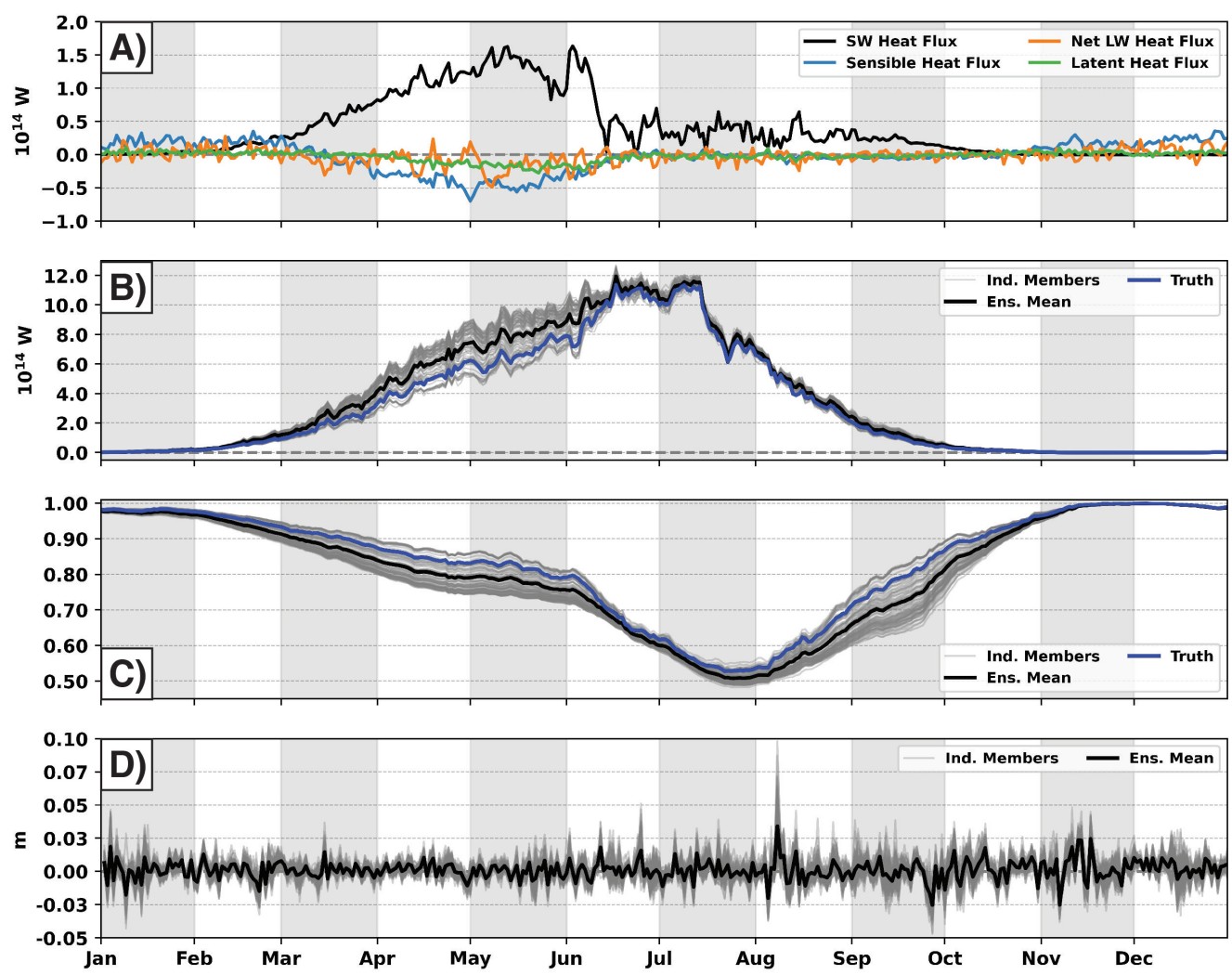

**Figure 11. Panel (A):** Ensemble mean daily biases of sea ice accumulated atmospheric heat fluxes for EAKF-ConcThick compared to the truth. The plotted atmospheric heat flux components include: shortwave heat flux (black line), sensible heat flux (blue line), net longwave heat flux (orange line), and latent heat flux (green line). Gray dashed line represents the zero reference line. **Panel (B):** Time series of sea ice accumulated shortwave heat flux for EAKF-ConcThick. The gray lines are the individual ensemble members, the black line is the ensemble mean, and the blue line is the truth. Gray dashed line represents the zero reference line. **Panel (C):** Same as Panel (B) but for mean surface albedo over sea ice. **Panel (D):** Daily biases of sea ice accumulated snowfall for EAKF-ConcThick compared to the truth. The gray lines are the individual ensemble members, and the black line is the ensemble mean.

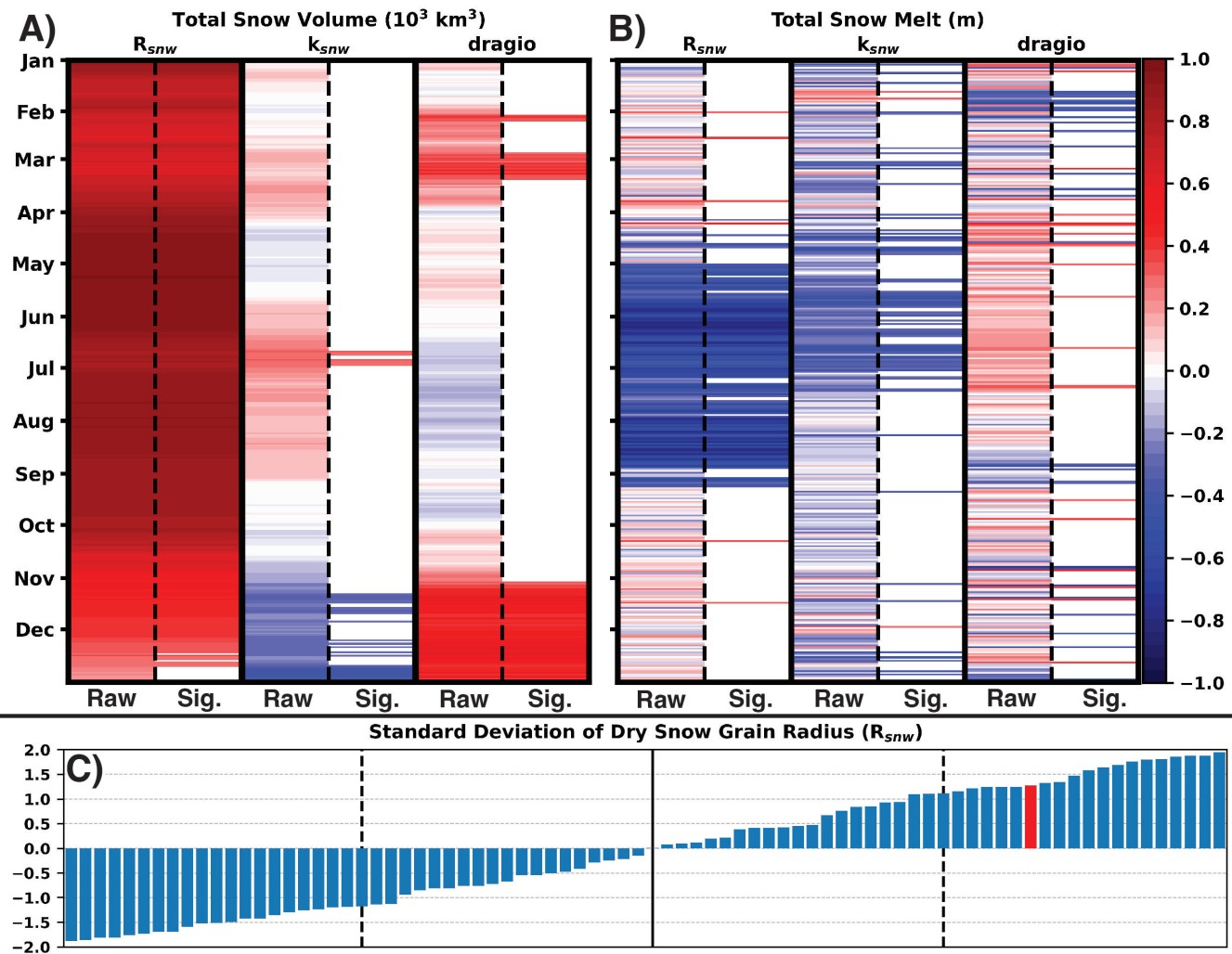

**Figure 12. Panel (A):** Daily correlations between perturbed CICE parameters and total snow volume over the Arctic. Correlations are computed using a Spearman's rank correlation method where both the raw correlations (**Raw**) and significant correlations with confidence at 99% (**Sig.**) are shown. **Panel (B):** Same as Panel (A) but for total snow melt over the sea ice in the Arctic. **Panel (C):** Sorted perturbed $R_{snw}$ parameter values for each ensemble member. Red bar indicates the truth member. Black line is the median and the two dash lines represent the interquartile range.