# Peer review of "Exploring Non-Gaussian Sea Ice Characteristics via Observing System Simulation Experiments"

_EGUsphere, 2023_

## Author Comment (AC1)

The authors are very grateful for the reviewer's time and effort in providing comments. Their work has greatly improved our manuscript. Our responses to each comment are included in black below.

**General Comments**

I wonder if even a data assimilation procedure that perfectly handled the non-Gaussian aspects of the problem would still result in a biased ensemble mean, since the distributions would necessarily be skewed. A discussion on what a "perfect" data assimilation procedure (i.e. something closely approximating Bayes theorem) would produce would be very helpful for the reader.

A sentence was added in the conclusion addressing this comment (line 405).

"This would include using distributions for the prior PDF and the observation likelihood that are similar to the observation error distribution and consider the bounds more appropriately (e.g., a truncated Gaussian distribution)."

A good example supporting the reviewer's hypothesis occurs when actual ice concentration is 0. Any ensemble estimate except for all 0's will be biased relative to the truth. However, it is less clear whether a perfect DA system can be unbiased compared to the observations. Since SIC is doubly bounded, generated observations would be inherently biased near either bound of zero or one. While the authors do agree that this would be helpful for the reader, it is a bit out of the scope of this work to speculate what a "perfect" data assimilation procedure would entail. The authors believe how to handle non-Gaussian situations is still an active area of research in the data assimilation community.

Also, I am interested to see if the ensemble median SIC is less biased than the ensemble mean and may provide a "better" measure than the mean for skewed distributions (though I admit the definition of "better" is not completely clear). This would not require additional experiments to be performed, just recomputing the bias scores by using the ensemble median, instead of the mean.

To address this comment the authors included several figures displaying the difference between using the ensemble median versus the ensemble mean. Figures 1 and 2 show the differences in the total Arctic sea ice area, sea ice volume and snow volume when using the mean versus the median (this is the same as figure 3 in the manuscript). The differences are quite small and not really noticeable. The same can be said about the spatial bias plots that were computed in the manuscript (Figures 3 and 4). The bias patterns and magnitudes are similar whether you use the median or the mean for computing the differences compared to the truth. The similarities between using the mean versus median is likely linked back to the ensemble spread being pretty small for the difference experiments and suggests that many of the distributions are approximately normal so that the mean and median are nearly the same. (see Figures 5 and 6 near the bottom of this document). The small ensemble spread likely means there are no extreme outliers effecting the computation of the mean. Due to the similarities, the authors have chosen to use the mean in this manuscript (line 185).

[Figure]

Fig. 1) Same as Figure 3 in the manuscript.

[Figure]

Fig. 2) Same as Figure 3 in the manuscript but using the ensemble median instead of the ensemble mean.

[Figure]

Fig. 3) Same as Figure 5 in the manuscript.

[Figure]

Fig. 4) Same as Figure 5 in the manuscript but using the ensemble median instead of the ensemble mean.

The previous paragraph concerning apparently biased distributions relates to a more general consideration of whether or not it makes any sense to evaluate a bias or using any other way of evaluating a single quantity, such as the mean, that has been extracted from the ensemble distribution. More general approaches, such as the continuous ranked probability score (CRPS), attempt to evaluate the accuracy of the entire ensemble distribution and not a single value obtained from the distribution. Please consider using such an approach for evaluating the resulting ensembles or explain why it hasn't been done.

Another reviewer mentioned adding an additional evaluation method that uses the ensemble instead of the mean. I chose to included the spatial probability score, which can be considered a spatial analogue of CRPS.

Line 65: "...if there is an optimal data assimilation setup...": This claim seems much too general, since the optimal configuration will likely depend on many details of a particular application of DA to sea ice. One such detail is the observing network, especially as mentioned elsewhere with respect to the unrealistic spatial distribution of SIC observations. Please rephrase this here and elsewhere.

This comment was addressed in the text.

line 65: "The OSSEs presented in this study will test different experimental setups to investigate their impacts on sea ice and snow states generated by data assimilation."

line 432: "These additional experiments would further help us understand the correct data assimilation setup for representing sea ice and snow in climate analyses."

Line 83: "One unique aspect of the EAKF...": This is misleading, since all variants of the ensemble Kalman filter use flow-dependent background-error covariances, not just the EAKF. Also, other data assimilation approaches, such as ensemble-variational approaches, use flow-dependent background-error covariances. Please rephrase.

This comment was addressed in the text.

line 83: "One important aspect of the EAKF is the ability to use a flow-dependent background-error covariance, which differs from a static background-error covariance typically employed by variational techniques."

Line 91: "...poor representation of model errors...": Presumably in an OSSE context you can perfectly represent any model errors that you choose to include in the experimental setup. Therefore, if inflation is still needed in such a context, then it must be serving a different purpose than accounting for model error. Please give some explanation for what these purposes are.

This was addressed in the text.

lines 90-94: "Adaptive prior covariance inflation was applied by "inflating" the prior background fields, increasing the variance by pushing ensemble members away from the ensemble mean (Anderson, 2007). Zhang et al. (2018) found a reduction in total Arctic sea ice area and volume errors when prior inflation was applied in their study."

Line 133: "...15% of the true values of SIC...": It is really 15% of the true values of SIC? Meaning that open water and very low concentration values are perfectly observed (i.e. error standard deviation close to zero)? Or is the standard deviation simply 15% (and not dependent on the true SIC)?

I added some text to the manuscript to make this comment more clear in the text.

line 133: "The observation error standard deviation for SIC is 15% of the true values of SIC (SICerror = SICtruth*0.15; Zhang et al. (2018)) and 0.1 m for SIT (approximation of future high precision data; Zhang et al. (2018))."

The observation error standard deviation is set by multiplying the truth SIC value by 15% ($SIC_{error}$ = $SIC_{truth}$*0.15). This method of specifying the SIC observation error follows the same method used in Zhang et al. (2018), which they state is "an approximate combination of bias and precision of the satellite-based concentration; e.g., Meier 2005." While our largest SIC uncertainty would be 15%, numerous other studies have employed other static or formula based methods for specifying the SIC observation uncertainty that would give similar values compared to our study (Tonboe and Nielsen, 2010; Mathiot et al., 2012; Zhang et al., 2021; Lee and Ham, 2022; Cheng et al., 2023). Additionally, this study was more focused on the data assimilation setup and not the observational side. However, the authors do agree that focusing on the observational setup (observation errors, observation density, etc) for data assimilation experiments should be addressed in future studies. Since other reviewers had comments on the SIC observation error specification, the authors tried to address some of the questions using the simplified data assimilation experiments that was presented in the paper (see section 3.3).

Line 134: "The locations for all synthetic observation types...": For all observation types? This is very unrealistic for SIC which is well observed almost everywhere by passive microwave satellite observations every day. CryoSat-2 is only used for ice thickness measurements. The retrieval process for obtaining ice thickness from these measurements also depends heavily on having accurate snow depth information, which is not well observed currently by any instruments and therefore the ice thickness measurements can have very high levels of uncertainty. I think that for this study to be relevant, a more realistic observing network must be used for all observation types (and also realistic values for observation uncertainties).

The authors addressed the SIC uncertainty in the comment above. The sea ice thickness uncertainty follows that used in Zhang et al. (2018), however, the authors realize that real sea ice thickness uncertainties can vary due to the different conditions. A sentence was included in the paper stating this.

line 135: "While studies that use real SIT observations have varied their uncertainties depending on their thickness value (Xie et al., 2018; Cheng et al., 2023), due to the complexity of computing SIT (Zygmuntowska et al., 2014) this study chose to use a single value for SIT uncertainty."

The authors did include a note in the text that our uncertainty value is for potential future datasets that might have higher precision. However, the sea ice thickness uncertainty chosen for this study is not too far off from values found in Zygmuntowska et al. (2014) (0.28 m in February/March and 0.21 m in October/November). In regards to the comment about the observing network, the authors chose the observing network for easy experimental setup and fair comparison between the observations that are assimilated in this study. A sentence was added in the text explaining why the network was chosen.

Line 136: The acronyms such as AICEN, VICEN, VSNON, SNWD, etc. are very non-intuitive and difficult to remember. These look like FORTRAN variable names, which are not necessarily appropriate for a scientific paper. Please consider variable names or non-acronym labels that readers will more easily remember and recognize. For example, if a quantity is a function of the thickness category, then this could be represented by a subscript of a variable corresponding to the thickness category index.

This comment has been addressed in the text. All references made to AICEN, VICEN, and VSNON have been replaced with category-based sea ice area, category-based sea ice volume, and category-based snow volume. Variables summed up over the different thickness categories are now referred to as SIC, $V_{ice}$, and $V_{snow}$. Observations are now referenced using SIC, SIT, $D_{snow}$, and SIST. I decided to leave the acronym for sea ice surface temperature (SIST) because it is close to the acronym for sea surface temperatures (SST) and easier to remember.

Line 226: "...not assimilating SIC observations improves most forecast metrics...": A more realistic (i.e. much denser) observing network for SIC would likely lead to more improvement when assimilating SIC since it would also lead to less ensemble spread and therefore reduced non-Gaussian effects. Also, as already mentioned, it's not clear if the observation error standard deviation for SIC observations is state dependent and, if so, if this could cause some of the resulting negative bias since low SIC observations will obtain more weight than high SIC observations.

As can be viewed below in Figures 5 and 6, the spread amongst the ensemble members for experiments 2 and 3 are quite small. This means the observations are having quite an impact on collapsing the initial spread that was found in the free forecasts (see Figure 1 in manuscript). The observation error specification was addressed in previous comments above. The authors believe that the underlying

issue is really related to the boundedness of the observation error distribution (bounded above 1 and below 0 for SIC) and the formulation of the EAKF and RHF using a Gaussian observation likelihood. This mis-match will lead to bias solutions which other studies have shown (see text for citations to these studies). Since SIC is doubly bounded, generated observations would be inherently biased near either bound of zero or one. Our observations would be more biased near 1 in our study since their observation errors would be large based on our specification. Even with higher SIC observation errors occurring for SIC values near 1 (over the central Arctic), the observations still have enough weight to pull our ensemble towards being negatively biased. The authors tried to address some of the questions regarding the observation error specification impacts on this study using the simplified data assimilation experiments (section 3.3).

[Figure]

Fig. 5) Daily total Arctic sea ice area, sea ice volume, and snow volume from experiment 2 where SIC, SIT and D$_{snow}$ observations are assimilated. Each gray line represents an individual ensemble member, black line represents the ensemble mean, and the red line represents the truth member.

[Figure]

Fig. 6) Daily total Arctic sea ice area, sea ice volume, and snow volume from experiment 3 where SIT and $D_{snow}$ observations are assimilated. Each gray line represents an individual ensemble member, black line represents the ensemble mean, and the red line represents the truth member.

**References**

Cheng, S., Chen, Y., Aydoğdu, A., Bertino, L., Carrassi, A., Rampal, P., and Jones, C. K.: Arctic sea ice data assimilation combining an ensemble Kalman filter with a novel Lagrangian sea ice model for the winter 2019–2020, The Cryosphere, 17, 1735–1754, 2023.

Lee, J.-G. and Ham, Y.-G.: Satellite-Based Data Assimilation System for the Initialization of Arctic Sea Ice Concentration and Thickness Using CICE5, Frontiers in Climate, 4, 797 733, 2022.

Mathiot, P., König Beatty, C., Fichefet, T., Goosse, H., Massonnet, F., and Vancoppenolle, M.: Better constraints on the sea-ice state using global sea-ice data assimilation, Geoscientific Model Development, 5, 1501–1515, 2012.

Meier, W. N.: Comparison of passive microwave ice concentration algorithm retrievals with AVHRR imagery in Arctic peripheral seas, IEEE Transactions on geoscience and remote sensing, 43, 1324–1337, 2005.

Tonboe, R. and Nielsen, E.: Global sea ice concentration reprocessing validation report, EUMETSAT OSI SAF Prod. Rep. OSI-409, version, 1, 18, 2010.

Zhang, Y.-F., Bitz, C. M., Anderson, J. L., Collins, N., Hendricks, J., Hoar, T., Raeder, K., and Massonnet, F.: Insights on sea ice data assimilation from perfect model observing system simulation experiments, J. Climate, 31, 5911–5926, 2018.

Zhang, Y.-F., Bushuk, M., Winton, M., Hurlin, B., Yang, X., Delworth, T., and Jia, L.: Assimilation of satellite-retrieved sea ice concentration and prospects for September predictions of Arctic sea ice, Journal of Climate, 34, 2107–2126, 2021.

Zygmuntowska, M., Rampal, P., Ivanova, N., and Smedsrud, L. H.: Uncertainties in Arctic sea ice thickness and volume: new estimates and implications for trends, The Cryosphere, 8, 705–720, 2014.

---

## Author Comment (AC2)

The authors are very grateful for the reviewer's time and effort in providing comments. Their work has greatly improved our manuscript. Our responses to each comment are included in black below.

**General Comments**

The biggest issue in the experiment design is the choice of "15% error for the true values of SIC", Line 133. Is this the right error model for SIC observations? SIC=0 and SIC=1 are two scenarios both with high accuracy in real data. Far away from the marginal ice zone, the observation and model should have good agreement on SIC and there is little need for any update. To reflect these scenarios, ensemble spread in SIC should be near zero for all the members have similar SIC values in those grid points, and for observation the error should also be near zero. However, 15% error means SIC=0 has zero error but SIC=1 has maximum error of 0.15. You mean SIC=0.99 observation can have an uncertainty bar of +-0.15?? This is certainly not realistic. In fact, I suspect that this specified observation error is causing all the problems in the following results (how assimilation of SIC degrade forecast, causing negative bias inside ice cover).

The authors have added some text to the manuscript to clarify our definition for specifying the SIC observation error clearer.

Line 133: "The observation error standard deviation for SIC is 15% of the true values of SIC (SICerror = SICtruth*0.15; Zhang et al. (2018)) and 0.1 m for SIT (approximation of future high precision data; Zhang et al. (2018))."

This method of specifying the SIC observation error follows the same method used in Zhang et al. (2018), which they state is "an approximate combination of bias and precision of the satellite-based concentration; e.g., Meier 2005." While our largest SIC observation error would be 15%, numerous other studies have employed other static or formula based methods for specifying the SIC observation error that would give similar values compared to our study (Tonboe and Nielsen, 2010; Mathiot et al., 2012; Zhang et al., 2021; Lee and Ham, 2022; Cheng et al., 2023). For this study, only sea ice concentration values greater than 0.01 are assimilated so that the observation error never reaches zero. A sentence in the text was added to make that clear.

Line 139: "Due to the SIC observation error method, only SIC observations greater than 0.01 (approximately the precision found in passive microwave sea ice concentration observation files, Meier et al. 2021) are assimilated."

The authors believe that the underlying issue is really related to the boundedness of the observation error distribution (bounded above 1 and below 0 for SIC) and the formulation of the EAKF and RHF using a Gaussian observation likelihood. This mismatch will lead to bias solutions that other studies have shown (see text for citations to these studies). Since SIC is doubly bound, generated observations would be inherently biased near either a bound of zero or one. Our observations

would have larger biases near one in our study since their observation errors are larger based on our specification. Even with higher SIC observation errors occurring for SIC values near 1 (over the central Arctic), the observations still have enough weight to pull our ensemble toward lower SIC values, leading to a negatively biased solution. During wintertime when the total sea ice area bias is small, the ensemble is smaller, which is visible in the free forecasts (see Figure 1 in manuscript). During the melt season and summertime, the ensemble spread in the total sea ice area increases. This increase in the spread allows the negatively biased observations to have more weight and pull the ensemble down to lower values. Since other reviewers had comments on the SIC observation error specification, the authors tried to address some of the questions using the simplified data assimilation experiments presented in the paper (see section 3.3). Additionally, this study was more focused on the data assimilation setup and not on the observational side. However, the authors agree that focusing on the observational setup (observation errors, observation density, etc) for data assimilation experiments should be addressed in future studies.

In reality, the model simulated SIC has most uncertainties in the marginal ice zone, the concentration elsewhere should be quite accurate. What's more uncertain is the thickness of ice and snow, and it is especially difficult to uncover the snow information from real satellite observations. Because altimetry will only measure the total freeboard, which is how much ice stick out of water level, it includes both ice thickness and snow depth. Other technique can infer snow depth information from heat signatures, but the observation has large uncertainties with a lot of assumptions made in the inference. Therefore a 0.1 m SIT error and 10% error for snow depth might be too optimistic. Thin ice is also not well represented in the model (given only category 1 is below 1 m), I'm not sure if uncertainty should be the same for thin (young) and thick ice. I suggest revisiting these uncertainties and checking against typical operational numbers.

The sea ice thickness uncertainty follows that used in Zhang et al. (2018), however, the authors realize that real sea ice thickness uncertainties can vary due to different conditions. A sentence was included in the paper stating this.

Line 135: "While studies that use real SIT observations have varied their uncertainties depending on their thickness value (Xie et al., 2018; Cheng et al., 2023), due to the complexity of computing SIT (Zygmuntowska et al., 2014) this study chose to use a single value for SIT uncertainty."

The authors did include a note in the text that our uncertainty value is for potential future datasets that might have higher precision. However, the sea ice thickness uncertainty chosen for this study is not too far off from values found in Zygmuntowska et al. (2014) (0.28 m in February/March and 0.21 m in October/November). The snow depth uncertainty follows that used in Rostosky et al. (2020), however, the authors realize that the snow depth uncertainties can depend on several different input parameters (e.g., brightness temperature, ice concentration, etc) and age of the sea ice the snow sits on (Rostosky et al., 2018). Rostosky et al. (2018) shows that snow depth uncertainties increase as the snow depth increases, which is similar to the observation error method used in our study. Since sea ice can exist and snow depth can be zero, the snow depth observation error was capped at a lower bound of 0.005 m. A sentence was added to the manuscript.

Line 140: "Similarly, the observation error for $D_{snow}$ has a lower bound of 0.005 m for observations that get close zero."

An additional motivation for lower snow depth uncertainties was to ensure that the observations impacted the background field. This helps illustrate the potential improvements that snow depth

observations could have if the uncertainties get small in the future. Once again, this study was more focused on the data assimilation setup and not on the observational side.

The definition of sea ice model and observed variables are quite confusing. There are "concentration" and "thickness", but later also "area" and "volume". My understanding is that SIC is equal to the integrated AICEN. Since volume = area x thickness, VICEN is actually not an independent variable from AICEN, but they are listed as separate variables in Table 1. The observation-state relation should be formulated in equations to be clear: how to compute SIT from AICEN (it's an average over 5 categories?) and SNWD is also confusing as how it relates to the model VSNON. There are at some point interchangeable use of the term "area" and "concentration", along with AICE and SIC. I suggest to define the state variables using easier-to-read math formula, such as $A_{ice,n}$ for AICEN and $h_{ice,n}$ for the thickness categories, then for observed quantities you can give equations such as $SIC = \sum_{n=1-5} A_{ice,n}$

The authors agree with this comment, and the manuscript was changed to address this comment. Starting at line 124, the descriptions and acronyms of the observations and the model state variables were changed. Formulas for computing SIC, SIT, $Dsnow$, and SIST were added to the document. In addition, the manuscript was changed to clear up any interchangeable us of the term area and concentration to ensure the right term is used.

Also, for the experiment names, it gets difficult to follow after the first three, you also keep repeating what each experiment does in the results section too. Can you provide more intuitive names, such as "Control", "EAKF-ConcThick", "Add-SnowD", and "RHF-ConcThick"?

The names for the different experiments are changed in the table and in the manuscript text.

Metric for forecast verification is chosen to be the total sea ice area and volume. These are integrated variables, so that local errors of positive and negative biases might get cancelled even before the error diagnostics. As you are performing an OSSE and you can evaluate against the truth member, why don't you choose a metric more suitable for the full 2D fields? You can even summarize errors separately for different areas (inside ice cover and near ice edge). At some point I was wondering if the ensemble spread is representing the correct amount of forecast errors for the ensemble mean versus the truth. It seems a measure for the skill of the ensemble is still missing in the current diagnostics. For sea ice, there are also a metric similar to the CRPS, called Spatial Probability Score (Goessling and Jung 2017), that you can try to add.

Another reviewer mentioned adding an additional evaluation method that uses the ensemble instead of the mean. I chose to included the spatial probability score. See Figures 3A and 7A.

**Line-by-line Comments**

**Line 45: "Common sea ice descriptive quantities":** Here you introduced concentration and thickness, but in the paper you are dealing a lot with area, volume and with respect to multi-thickness-category integrals. Can you provide some connection early on?

The authors added more details connecting the different observation quantities with the category-based state model variables in the paragraph starting on line 124. The authors feel that is the best place for this information since the introduction does not really discuss the model setup in terms of state variables that are updated.

**Line 65: "... if there is an optimal data assimilation setup...":** I would not use the word "optimal" here, since it is unclear whether the model or the observations are near the optimal condition to provide information about the true sea ice states.

Another reviewer commented on the use of the term "optimal." All references of "optimal data assimilation setup" have been removed from the manuscript.

**Line 77: "...follow closely to the optimal settings Zhang et al...":** Did you use exactly their setting or did you changed anything? Please describe more clearly. Again, I would refer the "optimal setting" as "setting that performed the best".

The authors modified the sentence to say "The CICE5 model setup" instead of "data assimilation settings." The data assimilation settings are modified for each experiment, which will be different compared to Zhang et al. 2018. Additionally, the first experiment labeled EAKF-ConcThick is an extension of the best performing experiment from Zhang et al. 2018. The differences are explained starting on line 157.

**Line 81: "Kalman 1960":** This is the reference for the Kalman filter, you should cite the EnKF reference (Burgers, or Evensen, or Carrassi's review, or even Houtekamer and Zhang review paper).

The sentence and citation have been changed on line 81.

**Line 83: "...accurate estimate...":** "best estimate" or "best guess" will be better here.

The sentence has been modified in the manuscript.

**Line 84: "flow-dependent background-error covariance":** You should point out that it is the ensemble that allows the estimation of a flow-dependent covariance, while the older methods such as "optimal interpolation" or 3DVar only employs static covariance.

The authors modified the sentence to make sure the readers know that the ensemble allows for estimating flow-dependent covariances.

**Line 86: "While the RHF...",** check grammar

Sentence was modified and moved to the future work portion of the conclusions.

Line 87: "however, that application is not applied in this study": Well, if not then you shall not mention this in the main methodology, you can discuss if this application is worth trying in future studies in the discussion, if that's what you are implying.
The authors agree with this comment. The sentence was moved to the future work portion of the conclusions.

Line 88: "horizontal localization": a more accurate term is "covariance localization", then you can comment that you only localize in the horizontal directions.
The sentence was modified using covariance location instead of horizontal localization.

Line 91: "inflating the prior background fields": its not the full fields that are directly inflated, rather the "ensemble perturbations" are inflated.
The sentence was modified to say that the ensemble perturbations are inflated, not the full fields.

Line 97: "sea ice thickness, which is represented by the product of sea ice volume and sea ice area": this is wrong...sea ice volume = area x thickness.
The word "product" was replaced with the word "quotient" to properly describe how sea ice thickness is represented in the sea ice model.

Line 99: "five categories with lower bounds of...": Why are these exact numbers chosen? Are you following some classic CICE configurations? I've seen other models using categories that are more focusing on thinner ice (0.1, 0.3, 0.7, 1.0, then 3.0), your values here seems to lean on the thicker categories.
The values for the five thickness categories follow what was used in Zhang et al. 2018. The thickness values are the original values when setting kcatbound = 0.

Line 100: "provides an unique challenge": "pose a challenge" will be more suiting.
The authors agree, and the sentence was modified.

Line 110: "non-Gaussian impacts": could you be more clear?
The sentence was modified so the statement is clearer for the reader.

Line 113: "three different parameters were perturbed that impact...": I would discuss the three parameters in separate sentences and be more clear. Please also list the range of values for the perturbation (one standard deviation around the true value), so that people can try to reproduce the experiments.

The authors have added additional discussion about the three parameters that were perturbed in this study. To make it easier for people wanting to reproduce our experiments, I have uploaded the perturbed parameter values to Zenodo for download. A link was included in the data availability section, and text was added to point readers to that section.

Line 114: "ocean-ice drag coefficient": The ice motion is mostly driven by atmospheric surface winds, why did you perturb the ocean drag coefficient instead? Wouldn't it be more responsive if you perturb the air-ice drag coefficients so that surface wind uncertainties can also be partially included in the ensemble perturbations?

The ocean-drag coefficient was chosen because it was one of the top parameters that provided uncertainty in CICE simulations in Urrego-Blanco et al. 2016. In addition, perturbing the ocean-ice drag coefficient can increase uncertainty in sea ice area and sea ice extent during winter according to Urrego-Blanco et al. 2016.

Line 116: "equilibrium state"? The model will experience seasonal cycles with the periodic forcings, how can it reach an equilibrium? I think you meant to spinup the model to reach the correct energetics, climatology?, or seasonal variability.

This sentence was modified in the manuscript.

Line 121: "negatively biased": Are all sea ice and snow variables negatively biased? Even for each thickness categories?

The authors do not believe that all sea ice and snow variables are negatively biased compared to the truth. Several ensemble members are greater than the truth value in sea ice area, sea ice volume, and snow volume in Figure 1. When comparing the ensemble mean, it is negatively biased for each of the sea ice and snow variables. The randomly chosen truth member is within the ensemble envelope during the entire year. Since there are individual members greater than the truth value, we believe that means the thickness categories are not all negatively biased.

Line 130: "sea ice and snow quantities have both ...bounds in their representation": "both" sounds weird here, how can a variable have both single and double bounds?

The sentence was modified in the manuscript to remove both.

Line 135: "10-second CryoSat-2": can you provide more details? Did you use 10 seconds worth of data amount from CryoSat-2?

The manuscript was modified to clarify this statement. The CryoSat-2 locations were recorded or measured every 10 seconds.

Line 136: "AICEN, VICEN, and VSNON": I don't think you have defined these names yet.
This sentence has been moved to the beginning of the paragraph. AICEN, VICEN, and VSNON have been redefined as well.

Line 141: "update the sea ice area in different categories": sea ice "area" and "concentration" seems to be used interchangeably, but it is confusing for the readers outside of sea ice community.
The authors tried to make it clear that the sea ice concentration is the observation and sea ice area is the model state variable that is updated by data assimilation. Additionally, the authors tried to make it clear in the manuscript how the sea ice concentration observation is related to the sea ice area model state variable (starting on line 127). The addition of the equation for how to compute sea ice concentration from the sea ice area model variable is also included.

Line 142: "allow sea ice conc. and volume to be updated": are the AICEN and VICEN two independent variables? I thought that VICEN = AICEN x hicen, and hicen is given (the five categories), so if you update concentration (AICEN) you already have the VICEN updated. Could you clarify?
$A_{ice,n}$ and $V_{ice,n}$ are individual variables in terms of output, however, they are computed by the equation listed in your comment. Two methods could be used for updating $A_{ice,n}$ and $V_{ice,n}$. A user could just update one of $A_{ice,n}$ and $V_{ice,n}$, and then update the other with the equation you have listed in the comment. This is the method chosen to update $V_{snow,n}$ in our experiment labeled EAKF-ConcThick. Zhang et al. 2018 chose to update both $V_{ice,n}$ and $V_{snow,n}$ using the data assimilation updated $A_{ice,n}$ and the equation in your comment. In our study, we saw that the updated $V_{snow,n}$ via post processing was just as good as the free forecasts of $V_{snow,n}$. This means the post process updates were not improving the representation of $V_{snow,n}$ compared to a free forecast. You are really depending on the updates of $A_{ice,n}$ from the data assimilation system to affect the volume fields. The other method is to allow the data assimilation system to update all three state model variables ($A_{ice,n}$, $V_{ice,n}$, $V_{ice,n}$). This method was applied in all of our experiments except the first one.

Line 145: In eq. 1, hsnon is defined as "category-based snow thickness". Are they given by the same 5 categories for sea ice thickness? It is not capital letter, which means they are not prognostic variables? How does this even work? Do you have hsnon scale with hicen (ice thickness categories)?
The different thickness category values listed in the method section under CICE are the lower bounds for each category. Once a thickness value goes over the category bound, the sea ice volume is transferred to the next category. The bounds for the five thickness categories are the same for sea ice and snow. I have added text in the manuscript to clarify this. The post-processing uses the prior sea ice thickness values in each category ($h^{prior}_{snow,n}$), which is computed by dividing the prior $V_{snow,n}$ by the prior $A_{ice,n}$ ($h^{prior}_{snow,n} = \frac{V^{prior}_{snow,n}}{A^{prior}_{ice,n}}$). The new snow volume is updated by multiplying the prior category-based snow thickness ($h^{prior}_{snow,n}$) by the updated category-based sea ice area ($A^{posterior}_{ice,n}$). The post-processing method was introduced in the Zhang et al. 2018 paper. Text was added to the manuscript to clarify the post-processing equations.

Line 147: for experiment 2, you are missing another important clarification here. Does all observation types (SIC, SIT and SNWD) update all model state variables (VICEN, AICEN, VSNON)? It maybe by default in DART that all cross-variable updates are kept, but for other systems there are often ad-hoc cross-variable localization. For example, not allowing SIT and SNOWD to change AICEN...so please carefully provide details. Table 1 doesn't have these information either, you need to state here more explicitly.

The authors agree with the reviewer's comment. Text was added in the DART section within the manuscript to mention that no cross-variable localization was applied in this study. Text was also added to clarify that while sea ice surface temperature observations are assimilated, sea ice surface temperatures in the different thickness categories are not updated by either post-processing or data assimilation.

Line 150: "non-parametric data assimilation": sounds like the new method for DA doesn't have parameters to tune...the non-parametric refer to the underlying distribution function. You should just say "non-Gaussian data assimilation"

Any uses of "non-parametric" when referencing the rank histogram filter were changed to "non-Gaussian."

Eqn. 2: this is not an equation, but a pseudo code. You can write AICEN ← AICEN * 1 / SIC.

The equation was modified in the manuscript.

Line 160-169: here you list three separate treatments for (1) SIC $>$ 1, (2) SIC within bounds, but AICEN $<$ 0, and (3) AICEN $>$ 0 but VICEN $=$ 0. In your code, what is the order for applying (1)(2)(3)? Does the order matter? For example, if I first get rid of negatives (2) then squeeze (1)? I'm not sure if the order matters.

The steps for the three different special treatments are laid out in Zhang et al. 2018. This study followed the same steps used in Zhang et al. 2018. During the experimental setup, early testing of the special treatments did not show that order matters.

Line 168: "mid-point sea ice thickness": It's not easy to understand that here you mean the thickness value for each category (bin), if you provide a SIT distribution (historgram over the 5 categories), before and after the three treatments, it would be much clearer.

The authors tried to make this statement clearer in the manuscript. The text now includes "the average thickness value allowed in the associated category (0.32, 1.01, 1.93, 3.51, 6.95)" where the values in parentheses are the values the sea ice area would be multiplied by to get the new sea ice volume in the category.

Line 175: "Total sea ice area, volume, snow volume": What's their connection to the model or observation variables? Can you provide equations for how you computed these integrated values?

Equations were included in the manuscript to help describe how these quantities are computed.

Line 179: "Total sea ice area will allow for better evaluation...": The reason is not clear to me.
This sentence has been modified due to another reviewer's comment.

Line 182: "Mean absolute bias and mean square error": Are they computed for the spatially-integrated values (total sea ice area, etc.) then averaged over time? Or are they also computed for the model states on each grid point, then taking both spatial and temporal averages? Again, equations may help a lot here for clarity.
Text has been added to the manuscript to clarify this comment. Additionally, equations were added to illustrate how MAB and MSE are computed in this study.

Line 190: "over-prediction, under-prediction": Could you define these please?
Text was added to the manuscript to clarify over- and under-predictions.

Line 198: "In experiments 1-3 investigate...": check grammar
This sentence has been changed.

Line 208: "the biases are reduced for total snow volume throughout": not quite reduced for winter seasons before Jun, what happened?
A couple of sentences were added at the end of the paragraph highlighting this finding and providing an explanation. Starting on line 253.

Line 214: "non-Gaussian component... could be the driving factor": from results of RHF I don't think the degradation in experiments 1 and 2 is due to non-Gaussianity. If so the RHF should have improved forecasts, but it in fact degrades even more. You should look for further reasons, such as ensemble spread compounded with the value bound, or strong seasonal cycles posing a large bias in the prior mean.
The authors believe that the underlying issue is really related to the boundedness of the observation error distribution (bounded above 1 and below 0 for SIC) and the formulation of the EAKF and RHF using a Gaussian observation likelihood. This mismatch will lead to bias solutions that other studies have shown (see text for citations to these studies). I have removed the word "non-Gaussian" from the sentence. Additionally, the authors tried to tie the SIC results to the fact that the SIC is doubly bound and often the SIC values reside near either bound.

Line 220: "removing the SIC observations provided a more accurate forecast": This might also suggest that the DA method is not causing the degradation, but rather the data themselves.
Since the SIC is doubly bound, the distribution that was used to generate the observations from the truth was a truncated normal distribution with bounds at zero and one. Thus, observations generated near the bounds will be inherently biased compared to the truth. If your truth value is 1, the randomly drawn observation will be biased just by the fact that the random draw cannot go above one. This means SIC observations that are randomly drawn from the truth near the bound of 1 will be biased low. I tried to address this issue with the simplified DA experiment where I changed the observation error values (section 3.3).

Figure 4: It seems most of the MSE is due to bias? MAB is almost equal to MSE. So maybe showing MAB is enough to tell the story.
The authors agree that MSE and MAB are almost equal. However, we chose to leave both for our evaluation in case readers are interested in seeing both.

Line 234: "total sea ice area ... performed poorly": based on your definition the SIC integral will include the negative biases, but if in operation people use the 15% concentration threshold to count for sea ice area (or extent) then this negative bias will not be a problem.
The biases would be more similar across the experiments if sea ice extent was used because the method does not multiply the grid cell area by the SIC value. Instead, the area is summed within the 15% SIC contour (according to the CICE source code). In fact, you see that for the SPS time series where the first three experiments are pretty similar throughout the cycling period. However, not accurately representing the actual SIC values over the main ice pack (SIC values come to 1) would lead to poor atmosphere-ocean-sea-ice feedbacks in a fully coupled model. Therefore, it is important to show that the total sea ice area is negatively biased and fully investigate why this low bias is present.

Line 244: "analysis increment (AI)": I would avoid using acronym that is already well-established.
The authors removed the use of the acronym AI from the manuscript.

Line 273: "drift away from true value towards the observed value": looks like the filter is putting too much weight on observation in this case? Does the truncation or the bound at 1 caused some error in observation uncertainty representation?

The simplified DA experiments were modified in the revised manuscript. However, the authors found that regardless of the filter type and observation error specification that the prior ensemble mean moves away from the truth. The authors believe that it is related to the observation error distribution that causes the problem and not the actual observation error value. In the simplified DA experiments, the truth value is very close to 1. Since SIC is bounded at 1 and 0, a truncated normal distribution is applied as the observation error distribution to generate the observations from the truth value and specified observation error. Since the truth is near 1, the observations will be low-biased compared to the truth. Even though each filter (EAKF and RHF) is formulated using different prior PDFs, the observation likelihood for both filters is assumed to be Gaussian. The mismatch between the observation error distribution and the observation likelihood weights the observations too much, leading to a low-biased solution for both filter types. The Gaussian likelihood has no knowledge of the bound at 1, however, the generated observations from the truncated normal distribution have the knowledge of the bound at 1 built in.

Line 286: "Applying a non-Gaussian ... can lead to erroneous observation impacts": If this is the reason for low biased analysis here, then applying RHF will reduce the bias? But you see the RHF actually increase the low bias in experiment 4... This is contradictory.

The authors response to this comment was answered in the above comment. The new simplified DA experiment in the manuscript tests both the EAKF and RHF with different observation error specifications. The RHF prior PDF is specified using a non-Gaussian histogram method, but the observation likelihood is still assumed to be Gaussian, which is similar to the EAKF. You see that the solutions for both filters in the simplified DA experiment are very similar, and the authors believe that is related to the fact that they both use a Gaussian observation likelihood.

Line 293: "near the sea ice margin": the area is called "marginal ice zone (MIZ)"

The manuscript has been updated using the term "marginal ice zone."

Line 298 and 329 and 336: "The use of non-parametric RHF did not handle the SIC obs better": again, the issue is not in handling the non-Gaussian distribution, but the observation error model is the culprit here. The true Bayesian solution might be low-biased SIC (given the observation), and RHF is actually doing a better job in fitting the observation, hence the "worse" low bias.

The response to this comment was answered in the reviewer's first general comments above in the document. If the true Bayesian solution is low-biased, which might be the case, this paper highlights the fact that data assimilation algorithms with Gaussian components will not give accurate solutions when compared to the truth.

Line 353: "forcing file for the truth member is an outlier": This is related to how you generated the perturbed members, if the forcing spread is large enough so that ensemble spread covers this truth member, then I would expect the filter to still work. Given here that the forcing has a near linear relation with the sea ice and snow states.

If the reviewer is referring to the CICE perturbed members, than the authors agree. Below the line for this comment, the manuscript discusses how users using the perturbed parameter method have to be mindful when evaluating results.

**References**

Cheng, S., Chen, Y., Aydoğdu, A., Bertino, L., Carrassi, A., Rampal, P., and Jones, C. K.: Arctic sea ice data assimilation combining an ensemble Kalman filter with a novel Lagrangian sea ice model for the winter 2019–2020, The Cryosphere, 17, 1735–1754, 2023.

Lee, J.-G. and Ham, Y.-G.: Satellite-Based Data Assimilation System for the Initialization of Arctic Sea Ice Concentration and Thickness Using CICE5, Frontiers in Climate, 4, 797 733, 2022.

Mathiot, P., König Beatty, C., Fichefet, T., Goosse, H., Massonnet, F., and Vancoppenolle, M.: Better constraints on the sea-ice state using global sea-ice data assimilation, Geoscientific Model Development, 5, 1501–1515, 2012.

Meier, W. N.: Comparison of passive microwave ice concentration algorithm retrievals with AVHRR imagery in Arctic peripheral seas, IEEE Transactions on geoscience and remote sensing, 43, 1324–1337, 2005.

Rostosky, P., Spreen, G., Farrell, S. L., Frost, T., Heygster, G., and Melsheimer, C.: Snow depth retrieval on Arctic sea ice from passive microwave radiometers—Improvements and extensions to multiyear ice using lower frequencies, Journal of Geophysical Research: Oceans, 123, 7120–7138, 2018.

Rostosky, P., Spreen, G., Gerland, S., Huntemann, M., and Mech, M.: Modeling the microwave emission of snow on Arctic sea ice for estimating the uncertainty of satellite retrievals, J. Geophys. Res. Oceans, 125, e2019JC015 465, 2020.

Tonboe, R. and Nielsen, E.: Global sea ice concentration reprocessing validation report, EUMETSAT OSI SAF Prod. Rep. OSI-409, version, 1, 18, 2010.

Urrego-Blanco, J. R., Urban, N. M., Hunke, E. C., Turner, A. K., and Jeffery, N.: Uncertainty quantification and global sensitivity analysis of the Los Alamos sea ice model, J. Geophys. Res. Oceans, 121, 2709–2732, 2016.

Zhang, Y.-F., Bitz, C. M., Anderson, J. L., Collins, N., Hendricks, J., Hoar, T., Raeder, K., and Massonnet, F.: Insights on sea ice data assimilation from perfect model observing system simulation experiments, J. Climate, 31, 5911–5926, 2018.

Zhang, Y.-F., Bushuk, M., Winton, M., Hurlin, B., Yang, X., Delworth, T., and Jia, L.: Assimilation of satellite-retrieved sea ice concentration and prospects for September predictions of Arctic sea ice, Journal of Climate, 34, 2107–2126, 2021.

Zygmuntowska, M., Rampal, P., Ivanova, N., and Smedsrud, L. H.: Uncertainties in Arctic sea ice thickness and volume: new estimates and implications for trends, The Cryosphere, 8, 705–720, 2014.

---

## Author Response (AR2)

The authors are very grateful for the editor's time and effort in providing comments. Their work has greatly improved our manuscript. Our responses to each comment are included in black below.

**General Comments**

I would like the authors to nuance their statement in the introduction that the best assimilation framework is not assimilating SIC and only assimilating snow. This seems in contrast to a vast amount of recent literature (i.e. on the importance of assimilating SIT), and is also showing a lack of understanding of the current lack of good snow on sea ice observations (they remain highly uncertain).

The authors have added text to the manuscript to address this comment.

**Line 9:** "Findings indicate that assimilating both sea ice thickness and snow depth observations while omitting sea ice concentration observations produced the best sea ice and snow forecasts, in our idealized experimental setup."

**Lines 46-71:** "The application of data assimilation to sea ice problems is not a novel idea since this research topic has been investigated for more than two decades. Common observation descriptive quantities for sea ice are concentration (e.g., the fraction of a grid cell covered with sea ice) and thickness (e.g., the sea ice surface extending down into the ocean). Previous studies have highlighted the importance of initial conditions when trying to predict Arctic sea ice from local to seasonal time scales, especially regarding accurate initialization of sea ice thickness (Msadek et al., 2014; Day et al., 2014; Dirkson et al., 2017). Although different data assimilation techniques have been used to update sea ice state variables (Meier and Maslanik, 2003; Van Woert et al., 2004; Lindsay and Zhang, 2006; Stark et al., 2008), numerous studies have tested updating sea ice state variables using the EnKF data assimilation method (Lisæter et al., 2003; Barth et al., 2015). These EnKF studies were tested both in a synthetic observation framework referred to as observing system simulation experiments (OSSEs; Barth et al. 2015; Kimmritz et al. 2018; Zhang et al. 2018) and using real observations from remote sensing platforms (Sakov et al., 2012; Massonnet et al., 2015). These studies found improvements in both sea ice analyses and their corresponding forecasts related to the spatial sea ice concentration field but little improvement in sea ice thickness. In addition, studies have improved the initialization of sea ice cover when updating sea ice thickness via a multivariate framework when assimilating only sea ice concentration observations (Massonnet et al., 2015; Sakov et al., 2012). More recent studies have tested the assimilation of sea ice thickness observations and found further improvements to both sea ice thickness and sea ice concentration states (Mathiot et al., 2012; Chen et al., 2017; Fritzner et al., 2018; Mu et al., 2018; Fiedler et al., 2022). While results from assimilating sea ice thickness observations are positive, they contain large observation uncertainties because satellite remote sensing retrieval algorithms contain large uncertainties due to input parameters and instrument errors (Kwok and

Cunningham, 2008; Zygmuntowska et al., 2014; Tilling et al., 2016; Xie et al., 2016; Ricker et al., 2017). Further research is needed to determine how to properly handle these uncertainties when assimilating sea ice observations. Lastly, there have been recent attempts to obtain observed snow depth from satellites; however, the uncertainties associated with these observations remain high (Maaß et al., 2013; Rostosky et al., 2018). Because snow is closely connected to albedo and sea ice melting, further understanding of the impacts of assimilating snow depth observations is needed. For example, Fritzner et al. 2019 found assimilating snow depth observations had positive effects on short-term forecasts of snow depth and sea ice concentration."

**Line 185:** "Since real world snow depth observations still have their limitations (Rostosky et al., 2018; Fritzner et al., 2019), the synthetic snow depth observations generated for this OSSE will test the impacts if high-quality snow observations are available year-round in the future."

**Lines 451-469:** "The first three experiments explore the impact of different assimilated synthetic observation subsets on the generation of the most accurate forecasts for both sea ice and snow states. According to the daily biases and aggregated statistics, EAKF-ThickSnow is more accurate, when compared to the truth, for sea ice area, sea ice volume, and snow volume. This highlights the negative impacts that SIC observations have on forecasts when they are assimilated in EAKF-ConcThick and EAKF-ConcThickSnow. This result contradicts previous studies that found positive impacts from assimilating SIC observations (Sakov et al., 2012; Massonnet et al., 2015; Posey et al., 2015). However, this result could be linked to differences in the observation error specification chosen for SIC observations in the different studies. In our study, early springtime SIC truth values are still close to one, maximizing their observation error (15% of the truth value), which leads to synthetic SIC observations being drawn further below the truth due to the bound at one. In addition, the prior spread increases because of the onset of springtime melt and prior inflation. Combining the low-bias observations with the increase in the prior spread leads to an enhancement of the non-Gaussian effects during early springtime. A similar but opposite effect (high-biased SIC observations) would be observed during winter; however, prior ensemble spread in the modeled SIC fields is smaller, resulting in a lower weighting of SIC observations. While potentially different from other studies, our chosen SIC observation error specification intensified the non-Gaussian effects of assimilating SIC observations while also showing the potential impact accurate SIT observations can have during data assimilation multivariate updating. Interestingly, SIC observations do provide positive updates in the marginal ice zone, as shown by SPS and total IIEE being lower in EAKF-ConcThick and EAKF-ConcThickSnow. Because of positive updates in the marginal ice zone, it would be optimal to assimilate SIC observations within the data assimilation system."

I am not totally satisfied as well that the authors have addressed the reviewers' comments regarding incorrect uncertainty attribution (0.1 for SIT is the current goal set for the future mission CRISTAL and is highly ambitious and by no means the current standard). I feel that the authors would do best to nuance their findings by adding 'in a idealised/synthetic/model data assimilation testing framework' or something along those lines.

The authors have modified the manuscript to ensure that the readers understand that this is an observing system simulation experiment (OSSE) that is being used as an "experimental" data assimilation framework to test different configurations.

**Line 6:** "This study presents different observing system simulation experiments (OSSEs), which through experimental observation networks and synthetic observations will provide a data assimilating testing framework."

**Line 162:** "The SIT observation error of 0.1 m is a goal for future satellite platforms and is not the observation error for current observing platforms."

**Line 73:** "Using OSSEs provides an experimental framework to test the impacts of synthetically generated observations in different data assimilation configurations."

Figure 2 is referring to real data location. Can you provide the data sources in the text. For panel (D) are you sure there are SST values along the CryoSat-2 tracks (?!). Again make it cristal clear (pun intended) to explain in the abstract, conclusion, text that your analysis is synthetic and doesn't use realistic data.
I have included the data source in the text.

**Line 167:** "The locations for all synthetic observation types that are assimilated were based on CryoSat-2 locations (locations measured every 10 seconds; more details on locations see CryoSat-2 Product Handbook at https://earth.esa.int/eogateway/documents/20142/37627/ CryoSat-Baseline-D-Product-Handbook.pdf), which provides the observational network for testing (Fig. 2)."

Caption for Figure 2 had been modified. Panel D is showing the locations of the synthetic sea ice surface temperatures (SISTs).

**Figure 2 Caption:** "A snapshot example of the spatial locations of the OSSE synthetically generated (A) sea ice area, (B) sea ice thickness, (C) snow depth and (D) sea ice surface temperature observations that are assimilated. The observation locations are from Cryosat-2 latitude and longitude ground tracks. Colorfill is the ensemble mean of the sea ice area and the dots are the observation locations along with their associated value."

I have included "synthetic" throughout the manuscript to be more clear that the observations that are assimilated in this study are not "real".

**Line 6:** "This study presents different observing system simulation experiments (OSSEs), which through experimental observation networks and synthetic observations will provide a data assimilating testing framework."
**Line 52:** "These EnKF studies were tested both in a synthetic observation framework referred to as observing system simulation experiments (OSSEs;"
**Line 70:** "Using OSSEs provides an experimental framework to test the impacts of synthetically generated observations in different data assimilation configurations."
**Line 151:** "Since satellites can not retrieve multi-category model quantities, aggregate synthetic observations are generated from the truth member to produce sea ice concentration (SIC), sea ice thickness (SIT),"
**Line 152:** "In this OSSE framework, synthetic observations are generated from the truth member using the forward operators and are assimilated."
**Line 155:** "This method was chosen to create the synthetic sea ice surface temperature observations

that were assimilated."

**Line 157:** "Because of this, we will use a single (SIT,$D_{snow}$) and double (SIC) truncated normal distribution when generating the synthetic sea ice and snow observations that are assimilated in our OSSEs."

**Line 165:** "Due to the SIC observation error method, only synthetic SIC observations greater than 0.01 (approximately the precision found in passive microwave sea ice concentration observation files, Meier et al. 2021) are assimilated."

**Line 167:** "The locations for all synthetic observation types that are assimilated were based on CryoSat-2 locations (locations measured every 10 seconds; more details on locations see CryoSat-2 Product Handbook at `https://earth.esa.int/eogateway/documents/20142/37627/CryoSat-Baseline-D-F pdf`), which provides the observational network for testing (Fig. 2)."

**Line 177:** "In EAKF-ConcThick, we allow the category-based sea ice area and volume to be updated independently by synthetic SIC and SIT observations while updating snow volume via post-processing."

**Line 185:** "In EAKF-ConcThickSnow, snow volume is no longer updated by post processing and assimilation of synthetic $D_{snow}$ is included in the assimilated observation subset."

**Line 88:** "To test the non-Gaussian effects of the synthetic SIC observations, EAKF-ThickSnow only assimilates synthetic SIT and $D_{snow}$ while allowing the category-based sea ice area, sea ice volume, and snow volume state variables to be updated from the observation increments."

**Line 196-198:** "Finally, EAKF-SIST tests the impacts of assimilating additional synthetic SIST observations to further improve the updates of sea ice and snow states. While synthetic SIST observations are assimilated, sea ice surface temperatures in the different thickness categories are not updated from the data assimilation step."

**Line 253:** "The first three experiments investigate which assimilated synthetic observation subset produces the most accurate forecasts for both sea ice and snow."

**Line 307:** "Evaluating analysis increments will help determine how the assimilation of synthetic SIC observations impact the different data assimilation experiments."

**Line 323:** "Even with a slightly higher IIEE, the removal of the synthetic SIC observations from the assimilate observation subset did provide better results."

**Line 326:** "The removal of SIC as an assimilated synthetic observation improved forecasts of total sea ice, however, forecasts of the sea ice edge were less accurate according to the total IIEE and SPS."

**Line 330:** "Three additional experiments were completed to investigate the impacts on sea ice when using a non-Gaussian RHF, modified forward operators for synthetic thickness observations, and the assimilation of synthetic SISTs."

**Line 369:** "With a constant truth value that does not change, synthetic observations are created that will be assimilated over the cycling period."

**Line 374:** "The experiments were cycled 5,000 times, assimilating the synthetic observations generated from the truth using a truncated normal distribution."

**Line 445:** "CICE-DART is used to conduct OSSEs to test different data assimilation configurations and the assimilation of different sea ice and snow observation subsets synthetically generated from a truth member."

**Line 448:** "The first three experiments explore the impact of different assimilated synthetic observation subsets on the generation of the most accurate forecasts for both sea ice and snow states."

**Line 454:** "In our study, early springtime SIC truth values are still close to one, maximizing their observation error (15% of the truth value), which leads to synthetic SIC observations being drawn

further below the truth due to the bound at one."

**References**

[revised manuscript text omitted]